# Tumor-targeted nanodrug FSGG/siGal-9 for transdermal photothermal immunotherapy of melanoma
Huihong Ren[1,2], Yujuan Zhang [3] ✉, Wei Huang[1], Haiyan Xu[1], Weixiong He[1], Nan Hao[1] & Cong Zhang[1]

Photothermal therapy (PTT) is a cancer-targeted treatment approach.The occurrence of tumors may be related to microbial infections (Viruses, bacteria, fungi, etc.), which probably provokes anti-tumor immunity. However, T cells in the context of cancer become exhausted and dysfunctional. Galectin-9 (Gal-9) is highly expressed in normal tissues and associates with body immune tolerance, and was firstly evidenced with much higher expression on the primary solid tumors than CD80/86 (B7) and CD274 (PD-L1) here, which suggests that Gal-9 may be a key factor in inhibiting the anti-tumor immunity, and its receptor T cell immunoglobulin and mucin domain 3 (TIM-3) was discovered on the cytotoxic T lymphocytes (CTL) with high expression as well based on the single cell analysis. The immune checkpoint communications showed that the Gal-9/TIM-3 axis played the most vital role on negatively regulating the anti-tumor immunity of CTL for melanoma. Then, we used a novel transdermal photothermal nanosensitizer (FSGG) loading Gal-9 siRNA (FSGG/siGal-9) for knocking the tumor cells down Gal-9 to block the Gal-9/TIM-3 axis and prohibit CTL exhaustion synergizing PTT against melanoma, which evidenced good effects on inhibiting tumor growth and enhancing anti-tumor immunity, named "photothermal immunotherapy". This paper provides a new perspective for tumor prevention and treatment.

Photothermal therapy (PTT) has become attractive for cancer therapy because of low invasiveness and high specificity and good therapeutic effects[1–3]. PTT can induce tumor cell apoptosis or necrosis or necroptosis, thus inhibiting tumor growth through localized hyperthermia[4–6]. Among various light absorbers in photothermal therapy, gold nanorod (GNR) is capable of variable longitudinal surface plasmon resonance (LSPR) and shows high efficiency in converting the photothermal effects of the near-infrared (NIR) spectrum that could penetrate into deeper tissues about minus 10 cm from the dermis[7]. It is widely used in PPT-based cancer treatments under in vivo studies[1–5].

The cause of tumor occurrence is mostly unknown, but the roles of microbe in cancer formation, diagnosis, prognosis, and treatment have been disputed for centuries. Recent studies have provocatively claimed that bacteria, viruses, and/or fungi are pervasive among cancers, key actors in cancer immunotherapy, and engineerable to treat metastases[8]. Our TCGA database analysis shows that tumors were significantly overexpressed microbial homologous proteins in common (For example MMP9, CDKN2A,

MYBL2)[9], which probably plays as tumor antigens provoking anti-tumor immunity and targets for chimeric antigen receptor T cell immunotherapy (Car-T therapy). However, in tumors, T cells are exposed to persistent immune checkpoints/co-inhibitory signals such as PD-L1/PD-1 (Programmed death ligand 1/programmed death 1), Gal-9/TIM-3 (Galectin-9/T cell immunoglobulin and mucin domain 3), HVEM/BTLA (Herpesvirus entry mediator/B and T lymphocyte attenuation factor), and CD155/TIGIT (CD155/T cell immunoglobulin and ITIM domain protein) pathways becoming exhaustion[10,11]. Exhausted T cells lose robust effector functions, express inhibitory receptors (IRs) including PD-1, TIM-3, BTLA, CTLA-4 (Cytotoxic T lymphocyte associated antigen 4), LAG-3 (Lymphocyte activation gene), TIGIT (T cell immunoglobulin and ITIM domain protein), and decline in the secretion of functional cytokines interferon-γ (IFN-γ), tumor necrosis factor-α (TNF-α) and interleukin-2 (IL-2)[12–15].

Our and others studies found that Gal-9 was highly expressed in normal tissues and the Gal-9/TIM-3 pathway associates with body immune tolerance[16]. Here, Gal-9 was firstly evidenced with much higher expression

[1]School of Pharmacy and School of Basic Medical Sciences, Nanchang University, Nanchang 330006, China. [2]Drug Dispensing Department, Zibo Central Hospital, Zibo 255000, China. [3]Department of Otolaryngology, The First Dongguan Affiliated Hospital, and Nursing College, Guangdong Medical University, Dongguan 523808, China. ✉e-mail: zhangyj@gdmu.edu.cn

on the primary solid tumors than such as CD80/86 (B7) and PD-L1 (CD274) as well, which suggests that Gal-9 may be a key factor in inhibiting the anti-tumor immunity. It is discovered that exhausted T cells are responsive to reinvigoration of immune responses by blockade of the PD-L1/PD-1 pathway[17]. For melanoma clinic treatment, two anti-PD-1 antibodies, nivolumab and pembrolizumab, were approved by the US FDA in 2014 for the treatment of metastatic melanoma but have been shown to produce objective response in just 30–40% of patients[18,19]. In 2019, studies discovered that Gal-9/TIM-3 pathway is the key mechanism of resistance to anti-PD-1 immunotherapy in lung cancer patients[20]. Gal-9 is recently defined as a multifaceted immune checkpoint that affects a host of cell types, such as altering macrophages to an anti-inflammatory, downregulating the number of effector T cells and increasing the number of regulatory T cells (Tregs) via triggering Gal-9/TIM-3 signaling pathway[13,21–23], making Gal-9 be a new potential target for immunotherapy.

We firstly explored the synergistic effects of PTT and silencing Gal-9 to achieve photothermal immunotherapy treating melanoma with a novel integrated nanostructure FSGG (Folic acid-modified and gold nanorod-linked iron oxide/silica core/shell nanostructure) /siGal-9 that is composed of gold nanorods, glucose-modified iron oxide-silica core/shell, and Gal-9-specific siRNA. Glucose modification can promote the cellular uptake via the glucose transporter families (GLUTs) that mediate the first step for cellular glucose usage and enable to sustain the energy demand required by cells for various biochemical programs. GLUTs has been upregulated in numerous cancer types including melanoma due to Warburg effect and are promising targets for the development of anticancer strategies[24–26]. The nanostructure FSGG/siGal-9 possesses comparative photothermal effects with GNR, proper size about ~111.67 nm diameter, good paramagnetic effects and hydrophilic surface that allows easily crossing the skin barrier, entering cytoplasm but blocked by nuclear pore (~70 nm[27], greatly reducing nuclear toxicity[28]), and eliminating from the body.

Besides recently, it is also revealed that heat (43 °C)-treated tumor cells are induced to release more exosomes that may interfere cancer immunotherapy since highly carrying such immune-related proteins as PD-L1, major histocompatibility complex class I (MHC-I), tumor-associated antigens (TAA) and so on[29–31]. Here, siGal-9 inhibiting immune checkpoint (Gal-9) in tumor cells and tumor-derived exosome may further promote therapeutic effects for photothermal immunotherapy.

## Results

### Significance of Gal-9 in tumor development

The formation, diagnosis, prognosis, and treatment of tumors may be related to microbial infections (Viruses, bacteria, fungi, etc.)[8]. Here, we analyzed the significantly higher expressed genes in the top ten type primary soild tumors including HNSC, KIRC, LUAD, LUSC, PRAD, THCA and UCEC from TCGA database (The Cancer Genome Atlas) as shown in Supplementary Table 1. The human tumors were significantly over-expressed microbial homologous proteins in common (31/55) as shown in Supplementary Table 1 via Align Peptide Mapping, especially for example *MMP9* (bacteria), *CDKN2A* (virus), *MYBL2* (fungi)[9].

Then, we studied the tumor microenvironment of melanoma using three samples for single cell analysis. The cell tpyes in tumor microenvironment (Fig. 1a) identified based on the markers in Supplementary Table 2. The PCA reduction dimension results, cells clusters and quality control data were supplied in the Supplementary Fig. 1. The cell communication based on the immune checkpoint in Fig. 1b indicated that Gal-9/TIM-3 is the key immune regulating axis on cytotoxic T cells in tumor microenvironment. Galectin-9 (Gal-9) is a β-galactoside binding lectin known for its immunomodulatory role as the ligand of T cell immunoglobulin and mucin domain 3 (TIM-3) which was originally found to be expressed on the surface of Th1 cells to induce cell apoptosis[32]. The results in Fig. 1c exhibited that *LGALS9* was higher expressed on the tumor cells (melanoma cells) and immune cells (such as T cells (cytotoxic T cells, exhausted T cells, Treg, T memory cells), B cells, NK cells and monocytes), but its receptor *HAVCR2* was relatively much higher expressed on the

cytotoxic T cells and NK cells which have been evidenced as the main tumor killers in vivo[11,33]. Therefore, Gal-9 may be as a new target for melanoma immunotherapy.

Since the Gal-9/TIM-3 pathway was also newly found associating with body immune tolerance[16]. Therefore, we checked the expression levels of Gal-9 and TIM-3 in normal tissues and organs in Supplementary Fig. 2, which are much higher than the expression level of forkhead box P3 (*FOXP3*) always as the master transcription factor of regulatory T (Treg) cells maintaining immune tolerance[34]. To evidence the possibilities of the Gal-9/TIM-3 pathway for cancer immunotherapy, we also used bulk rna_seq data from the TCGA database for bioinformatics analysis, and the results shows that the expression level of *LGALS9* (Gal-9) is much higher than immune checkpoints such as CD80/86 and CD274 (PD-L1) in the top type tumors (Supplementary Fig. 3) including TCGA-SKCM (malignant melanoma) as shown in Fig. 1d, meanwhile positive correlation with tumor size (Supplementary Fig. 4). Here, all the data suggest that Gal-9 may be a key factor in inhibiting the anti-tumor immunity as well.

### Synthesize and characterization of FSGG/siGal-9

Gold nanoparticles normally exhibit high photothermal conversion efficiency and easily synthesize. First, we successfully constructed a new multifunctional composite nanocarrier FSGG, which mainly contains the following steps (Illustration in Materials and Methods): firstly, $Fe_3O_4$ was used as the core and TEOS was added to form porous $SiO_2$ coat on $Fe_3O_4$ surface for building $Fe_3O_4@SiO_2$ (FS); secondly, PEI-PEG-Glu was used to modify the surface of FS to build $Fe_3O_4@SiO_2@PEI-PEG-Glu$ (FSG), as shown in Fig. 2a; finally, GNR was synthesized by seed growth solution method and modified with MUA, and then assembled on the FSG surface by electrostatic adsorption to achieve $Fe_3O_4@SiO_2@PEI-PEG-Glu@GNR-MUA$ (FSGG) (Fig. 2b). In addition, the FSG nanoparticle size distribution was uniform about ~91.67 nm, and GNR-MUA (Average length about ~35 nm and average width about ~10 nm, and the aspect ratio about ~3.5, Supplementary Fig. 5) was uniformly adsorbed on the FSG surface, and the nanoparticle dispersion was good (Fig. 2a, b).

The Zeta potential of FS, FSG, GNR-MUA, and FSGG were $-33.13 \pm 2.15$ mV, $+49.6 \pm 1.82$ mV, $-32.77 \pm 0.71$ mV, $+29.97 \pm 1.21$ mV, respectively as shown in Fig. 2c. The changes of zeta potential indicated that composite nanomaterials are successfully synthesized via layer by layer.

The UV-Vis absorption spectrum (Fig. 2d) showed that the plasmon resonance absorption peak of GNR-MUA was 768 nm, while that of FSGG was 784 nm, resulting in 16 nm redshift of the peak, which implied that the composite nanomaterials were very good retain the optical properties of GNR-MUA.

After synthesis of the FSGG constructure, we tested the capacity of the nanoconstructure to carry the Gal-9 siRNA (siGal-9) via electrostatic interactions between PEI (High positive charge) on the FSGG and the siRNA (Negatively charged). A gel shift assay was used to evaluate and corroborate the capacity of FSGG binding to siGal-9. Since free siRNA migrates faster on agarose gel, whereas the siRNA bound to FSGG is apparently immobile, the gel shift was commonly used to visually assess binding capacity. The results of gel shift in agarose showed that when the mass ratio of FSGG to siRNA (w/w) was higher than 6:1, the siRNA was maxumuly loaded (Fig. 2e).

It has shown that gold nanorods (GNR) are promising candidates for photothermal therapy under NIR irradiation. Therefore, we determined the photothermal effects of FSGG under the 808 nm laser irradiation at 2 W/cm² of optical density for 20 min. As shown in Fig. 2f, GNR-MUA (40 μg/ml) was rapidly heated to 48 °C, and FSGG (40 μg/ml) also increased the temperature to 48 °C after 5 min of laser irradiation. When prolonging the irradiation time to 20 min, GNR-MUA (40 μg/ml) enhanced the temperature to 59 °C, and FSGG (40 μg/ml) raised the temperature to 57 °C, which exhibits GNR-MUA and FSGG possessing comparative photothermal efficiencies and obviously exceeds the threshold temperature for hyperthermia killing the tumor cells. Here, we calculated the photothermal conversion efficiency (PCE) ξ of 40 μg/ml GNR-MUA (25.1%) and

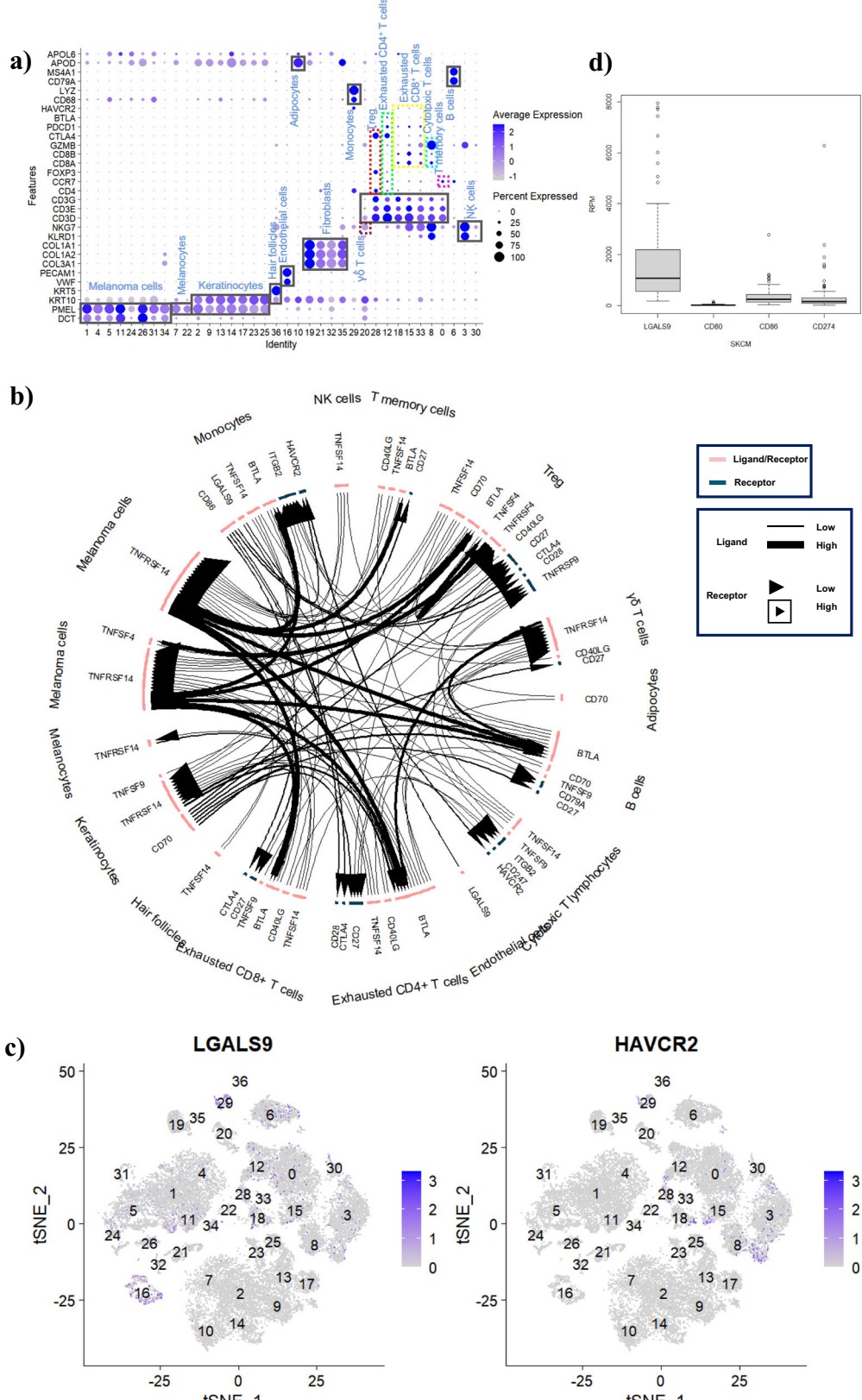

**Fig. 1 | LGALS9 (Gal-9) and melanoma. a** Cell types in tumor microenvironments identified based on the markers in Supplementary Table 2. **b** Immune checkpoint communications on the cells in the tumor microenvironments. **c** LGALS9 (Gal-9) and HAVCR2 (TIM-3) expressed in the tumors. **d** Expression level of LGALS9 (Gal-9), CD80/86 and CD274 (PD-L1) in primary soild SKCM (Skin cutaneous melanoma) tumors from TCGA (The Cancer Genome Atlas).

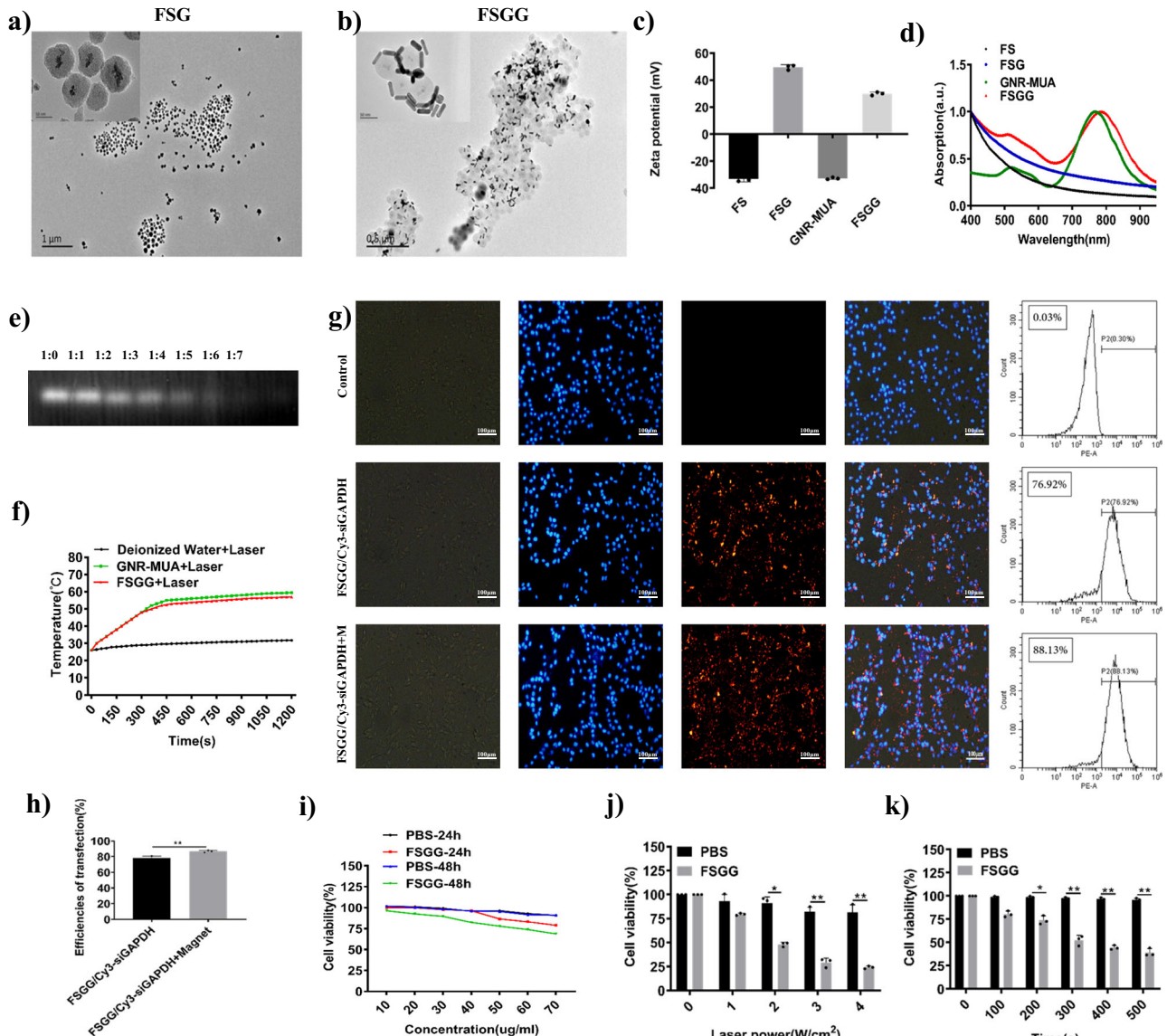

**Fig. 2 | Synthesize and characterization of novel multifunctional nanostructure FSGG/siGal-9. a, b** TEM images of FSG and FSGG. FSG and FSGG were synthesized as described in Materials and Methods. **c** Zeta potential and (**d**) Normalized UV-Vis absorption spectra of FS, FSG, GNR-MUA, FSGG, which were synthesized as described in Materials and Methods. **e** Gel shift assay for assessing the siRNA-loading capacity of FSGG. Equal volumes containing 1 μg of siRNA and the desired weights (0 μg, 1 μg, 2 μg, 3 μg, 4 μg, 5 μg, 6 μg, 7 μg) of FSGG were mixed and incubated at the indicated weight ratios for 30 min. Gels were visualized with Ethidium Bromide. **f** Temperature increase induced by photothermal effects of GNR-MUA or FSGG in vitro. 1 ml of GNR-MUA (40 μg/ml) or FSGG (40 μg/ml), or deionized water was irradiated by 808 NIR laser at 2 W/cm2 for 10 min. The temperatures were measured using infrared radiation thermometer and plotted at indicated time points. **g, h** Cellular transfection efficiency of siRNA by FSGG in vitro. After 24 h incubation, the intracellular fluorescence intensity was observed under fluorescence microscope (**g**) (The cell nuclei were stained with DAPI, scale bar =

100 μm) and the Cy3 positive population of B16-F10 was determined by flow cytometry (**h**). **i** Assessment of cytotoxicity of FSGG in vitro. B16-F10 cells were incubated with a cell culture medium containing various concentrations (10 μg/ml, 20 μg/ml, 30 μg/ml, 40 μg/ml, 50 μg/ml, 60 μg/ml, 70 μg/ml) of FSGG or PBS as control for 24 h or 48 h. The cell viabilities were measured by the MTT assay. **j, k** In vitro photothermal effects of FSGG killing B16-F10 tumor cells. B16-F10 cells were incubated with 40 μg/ml FSGG or PBS overnight. Subsequently, the cells were irradiated by 808 nm NIR laser for 5 min at various densities (0 W/cm², 1 W/cm², 2 W/cm², 3 W/cm² and 4 W/cm²) (**j**), or at 2 W/cm² for various times (0 s, 100 s, 200 s, 300 s, 400 s, 500 s) (**k**). After 24 h, the cell viabilities were measured by MTT assays and the percentages of dead cells = (Cell viabilities before laser irradiation - cell viabilities after laser irradiation)/ (cell viabilities before laser irradiation) were calculated. Error bars represent the standard deviation of 3 experiments (*$P \leq 0.05$; **$P \leq 0.01$; ****$P \leq 0.0001$; Adding the magnet field is denoted with "+M").

40 μg/ml FSGG (22.8%) via the below equation using the 24-well polystyrene plate at room temperature (The explanatory details in the Supplementary Materials 1: Supplementary Note 1–3, Supplementary Figs. 6, 7 and Supplementary Table 3):

$$\triangle T_{max} = \frac{\xi I S d}{\lambda A} \qquad (1)$$

where $\Delta T_{max}$ is the maximum temperature difference on both sides of thin wall, i.e., heat conduction driving force (°C), I is the light intensity (2 W/cm²), S is the light spot area (0.384 cm²) irradiated to the sample, d is the thin wall thickness (0.007 m), A is the area of thin wall ($3.256 \times 10^{-4}$ m²), λ is the thermal conductivity (W/(m·°C)), the difference is small within a certain temperature change, which is close to a constant, such as transparent polystyrene material λ is 0.15 W/ (m·°C) at 60 °C.

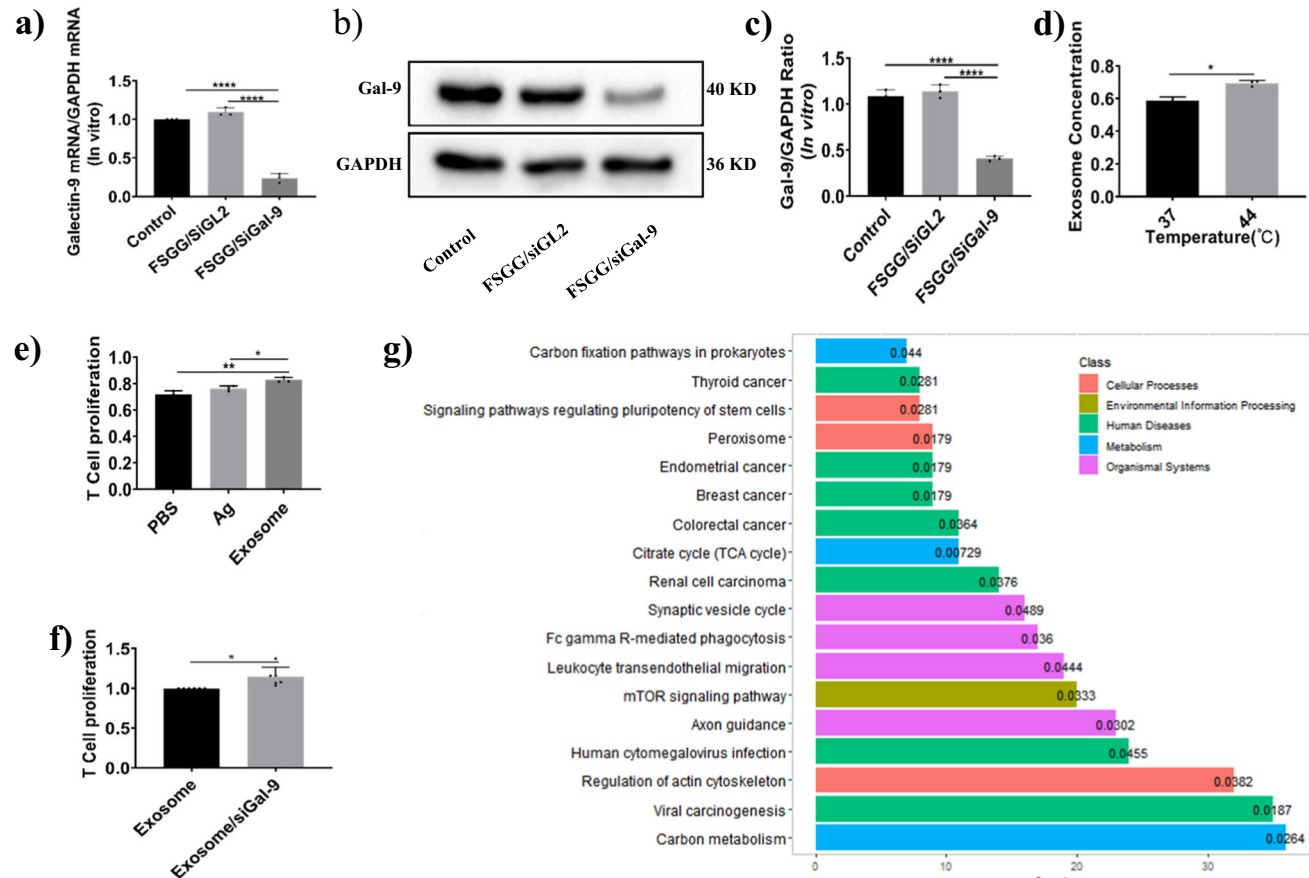

**Fig. 3 | Gene silencing by FSGG/siGal-9 in vitro and biological functions of exosome/siGAL-9.** Gene silencing efficiency using FSGG/siGal-9 in vitro. The mRNA expressions of Gal-9 and GAPDH were quantified with RT-q-PCR (**a**), or by western blots (**b**) and (**c**). **d** Changes of exosome concentration released by B16-F10 cells at different culture temperature 37 °C or 44 °C for 30 min. The concentration was quantified with protein levels which was examined by the BCA method. **e, f** Exosome and exosome/siGal-9 stimulating tumor-specific splenic lymphocyte proliferation. Exosome was collected from the culture medium of B16-F10 cells, while exosome/siGal-9 was collected from the culture medium of Gal-9-silenced B16-F10 cells. Antigen (Ag) was directly prepared from B16-F10 cells via the frozen-thawed method. The BCA method was used to quantify the concentration of Ag, exosome and exosome/siGal-9. The same amount (50 μg/ml) of Ag, exosome or exosome/siGal-9 was added into the cultured tumor-specific splenic lymphocytes for 24 h. Then the CCK-8 method was used to detect the proliferation of lymphocytes. **g** Proteomics of exosome or exosome/siGal-9 were analyzed by LC-MS/MS, and the main biological function changes of exosome/siGal-9 were depicted according to the KEGG pathway where bars were labeled with *P* value. For the KEGG pathway, the crit-up ratio is ≥1.2 and the crit-down ratio is ≤1/1.2. In total, 379 proteins were up and 746 proteins were down. Error bars represent the standard deviation of 3 experiments (*$P \leq 0.05$; **$P \leq 0.01$; ****$P \leq 0.0001$).

Next, we tested the cellular delivery efficiencies of siRNA by FSGG. The upregulation of GLUTs has been reported in numerous cancer types including melanoma. It was observed that the successful delivery of siRNA into the cytoplasm by FSGG/siRNA or FSGG/siRNA+Magnet, compared to PBS/siRNA control was evidenced under fluorescent microscope via Cy3-labeled siRNA and DAPI-stained nuclei (Fig. 2g). Besides, FSGG nanostructure contains the magnetic core, so magnet field has also significantly enhanced cellular delivery efficiencies of siRNA by FSGG/Cy3-siGAPDH in Fig. 2h (Cellular delivery efficiencies: 88.13% by FSGG/Cy3-siGAPDH+M v.s. 76.92% by FSGG/Cy3-siGAPDH).

In order to assess the biocompatibility of this novel FSGG nanostructure, we tested cellular toxicity of FSGG with various concentrations at different incubation time. Here, the results showed that the cell survival rate was as high as 78.91% in B16-F10 cells after incubation with FSGG at a concentration of 70 μg/ml after 24 h; while the cell survival rate were decreased to 68.76% when the incubation time was extended to 48 h. However, more than 82.46% of the cells survived after 48 h incubation at 40 μg/ml of FSGG, showing low cytotoxicity (Fig. 2i). Therefore, we chose the 40 μg/ml of FSGG for further experiments. Taken together, these data suggested that the newly synthesized nanostructure possesses good biocompatibility.

To further explore the killing tumor cell capacity by the photothermal effects of FSGG in vitro, we incubated B16-F10 cells with 40 μg/ml FSGG in 96-well plate and then irradiated with 808 nm near infrared laser. The MTT results showed that the photothermal effects of FSGG exhibited a significant tumor cell killing ability when irradiated under 2 W/cm² for 5 min (Fig. 2j) or after 200 s at 2 W/cm² (Fig. 2k), compared to the control cells incubated with PBS under the same irradiation conditions. All these results present that the photothermal effects of FSGG could effectively inhibit tumor cell viability in vitro.

## Gene silencing by FSGG/siGal-9 in vitro and exosome/siGAL-9 functions

Here, we tested the ability of FSGG/siRNA to silence the targeted Gal-9 gene in melanoma cells. The gene expression of Gal-9 in B16-F10 cells was significantly repressed, by nearly 76.15 ± 5.77% at mRNA level (Fig. 3a) and, 59.29 ± 2.63% at the protein level (Fig. 3b, c) after transfection with FSGG/siGal-9.

Besides, heat (43 °C)-treated tumor cells were induced to release more exosomes that may interfere cancer immunotherapy since highly carrying such immune-related proteins as PD-L1, MHC-I, TAA and so on. We also found that the 44 °C apparently induced the B16-F10 cells to release more

exosomes than 37 °C (Fig. 3d). Meanwhile, we also compared the stimulating capacities of tumor antigen (Ag), exosome and exosome/siGal-9 to the tumor-specific T lymphocyte proliferation. As shown in Fig. 3e, f, exosome possesses higher stimulating capacity than Ag, while Gal-9-silenced exosome (exosome/siGal-9) further enhanced this stimulating capacity to $1.15 \pm 0.050$ compared to exosome. Then, we performed the proteomic analysis of Gal-9-silenced exosome. As shown in Fig. 3g, most of proteins were distributed among the following aspects: carbon metabolism, viral carcinogenesis, regulation of actin cytoskeleton, human cytomegalovirus infection, axon guidance, mTOR signaling pathway, leukocyte trans-endothelial migration Fc gamma R-mediated phagocytosis, revealing the role of exosome/siGal-9 in the enhancement of the stimulating capacity of T lymphocyte proliferation.

### Immunotoxicity and toxicity of FSGG for topical application

Transdermal drug delivery nanosystems possesses several advantages such as reducing associated systemic toxicity or side effects, prolonging drug stabilities, controlling drug releases[35]. Previous studies exhibit that hundreds of nanometers of nanocarriers could effectively pass through the skin barrier for drug delivery[36]. Studies also showed that particle size and surface characterization mainly affect their biocompatibility, in vivo distribution and elimination[28]. In humans, the nuclear pore complex (NPC) has an outer diameter of ~1200 Å, an inner diameter of ~425 Å for the central transport channel, a height of ~800 Å[27]. Thus, the size of drug nanocarrier is larger than the diameter of nuclear pore, which is retarded to enter the nuclei, and finally may be better biocompatibility and faster elimination. Here, we assess the in vivo immunotoxicity and toxicity of the composite

FSGG (The FSG nanoparticle size distribution was ~91.67 nm (Fig. 2b), and the average length and width of GNR-MUA were ~35 nm and ~10 nm, respectively (Supplementary Fig. 5)) for topical administration. All the mice were massaged with the preparations of FSGG plus magnet field every 3 day for the treatment period of 22 days on the right back flank skin where the tumor would be inoculated. And then we sacrificed the mice after the 22 days or 3 months from the starting. Firstly, we tested the percentage of $CD4^+$ and $CD8^+$ T lymphocytes in the spleen as the largest peripheral immune organ. The results (Fig. 4a–d) showed that compared with the PBS-treated control group, the percentage of splenic $CD4^+$ and $CD8^+$ T lymphocytes in the FSGG-treated mice did not change significantly no matter 22 days or 3 months. Secondly, in order to assess whether the composite nanomaterials affect the main organs of mice, the levels of GPT, GOT, CRE and BUN in serum were tested. And the histopathology of tissues from the liver, spleen, lung, and kidney of mice were harvested and examined. All the results exhibited that the FSGG treatment had no obvious hepatotoxicity and nephrotoxicity in mice (Fig. 4e–h). Moreover, we did not observe any histopathological abnormalities in any of the mouse organs as well (Fig. 4i). Furthermore, we monitored mice every day for 3 months and did not observe any clinical signs of toxicity, including ruffled fur, impeded movement, signs of abnormal constitution, aberrant behavior, loss of weight, ocular or nasal discharge, respiratory distress, inability to walk, or diarrhea. The above results indicated that the novel nanostructure FSGG has no obvious immunotoxicity and toxicity during the application period (22 days) in vivo.

Additionally, we detected the accumulation of FSGG in liver and spleen after the topical administration of 22 days using TEM. As shown in Fig. 4j, there is no obvious accumulation of FSGG, or FSG, or GNR-MUA in liver and spleen after the topically transdermal administration of FSGG (Controls were played in Supplementary Fig. 8). All the data manifest that the 22-day topical administration of FSGG preparation is save in the mouse model.

### Photothermal-immunotherapy for melanoma using FSGG/siGal-9

Then, we assess in vivo tumor-targeted siRNA delivery by FSGG in vivo. To determine the in vivo transdermal delivery efficiencies of FSGG/siRNA into tumor, we applied Cy3-labeled FSGG/siRNA on the top of tumor tissue in the B16-F10 melanoma-bearing mice. The results showed that FSGG/Cy3-

siRNA in magnet field showed the highest efficiency in delivering siRNA into tumors, as compared to control mice treated with PBS/Cy3-siRNA or GNR-FA/Cy3-siRNA or FSGG/Cy3-siRNA without magnet field (Fig. 5a). Therefore, we applied the FSGG/Cy3-siRNA plus magnet field for the following animal studies.

Next, we evaluated photothermal effects induced by FSGG in vivo. We topically treated the B16-F10 melanoma-bearing mice with FSGG in the presence of magnet field, and then irradiated the mice by NIR laser and monitored the temperature by an infrared thermometer. As shown in Fig. 5b, FSGG greatly increased the temperature of the treated zone to 45.4 °C compared to PBS control (37.1 °C) after 5 min of the NIR laser irradiation, exhibiting an excellent photothermal effects in vivo.

Immune checkpoint blockages such as programmed death-1 (PD-1) antibodies has been emerging anti-tumor treatment for metastatic melanoma, however the objective response only achieved 30–40% in patients. In 2019, studies discovered that Gal-9/Tim-3 pathway is the key mechanism of resistance to anti-PD-1 immunotherapy in cancer patients[20]. Our results in Fig. 1b firstly found that Gal-9/TIM-3 may be the key reason for exhausting cytotoxic T cells in the acral melanoma microenvironment. Previous studies evidenced that cytotoxic T cells played the main role in anti-tumor immunity[11]. Here, we explored the combined therapeutic regimen using FSGG-siGal-9 siRNA that exerts dual photothermal effects and knocking down Gal-9 on melanoma in vivo. As shown in Fig. 5c, the tumor grew aggressively in the mice sham-treated with PBS with or without irradiation; the treatment with non-specific siRNA-conjugated FSGG ("FSGG/siGL2+M") only achieved minor suppression of tumor growth, however, the treatment with FSGG/siGal-9 ("FSGG/siGal-9+M") significantly delayed the tumor growth by day 22 post-inoculation of tumor cells, which is consistent with the varieties of tumor weight after the corresponding treatments in Fig. 5d. The Gal-9 gene silencing efficiencies were confirmed by RT-q-PCR results as well in Fig. 5e. These results certified that blocking the Gal-9/TIM-3 axis could significantly inhibit melanoma growing.

Besides, the treatment with FSGG/siGL2 plus laser irradiation ("FSGG/siGL2+Laser+M") that induced the photothermal effects in vivo also significantly delayed the tumor growth, as compared to the treatment with control treatment using PBS plus laser irradiation or FSGG/siGL2 ("FSGG/siGL2+M") (Fig. 5c, d). To further confirm the photothermal effects in vivo, we monitored the temperature of irradiated zone during the entire treatment period using an infrared thermometer. As shown in Fig. 5f, the treatment with FSGG/siGL2 plus laser irradiation plus magnet field ("FSGG/siGL2+Laser+M") or FSGG/siGal-9 plus laser irradiation plus magnet field ("FSGG/siGal-9+Laser+M") significantly increased the temperature to $42.13 \pm 0.68$ °C or $44.93 \pm 0.42$ °C, respectively, compared to the treatment with PBS plus laser irradiation ("Control+Laser", $36.87 \pm 0.21$ °C).

The anti-tumor effects were highest when the melanoma-bearing mice received a combination treatment with FSGG/siGal-9 plus laser irradiation plus magnet field ("FSGG/siGal-9+Laser+M"), which further suppressed tumor growth (Fig. 5c) and also was confirmed by the tumor weight (Fig. 5d) that was remarkably smaller ($0.49 \pm 0.05$ g) than that in the mice receiving single treatment with the photothermal effects ("FSGG/siGL2+Laser+M", $0.86 \pm 0.10$ g) or gene silencing of Gal-9 ("FSGG/siGal-9+M", $0.76 \pm 0.04$ g). All the gene silencing efficiencies and the photothermal effects were confirmed in Fig. 5e, f.

### FSGG/siGal-9-mediated photothermal-immunotherapy enhancing anti-tumor immunity

Studies discovered that Gal-9 expressed by tumor cells or APCs (DCs) induced T cell apoptosis via Galectin-9/TIM-3 pathway[13,22,23]. Dendritic cells (DCs) are accepted as the only dedicated antigen-presenting cells (APC) in the immune system, which can internalize, process and present antigens (Ag) to naive T cells and subsequently activate them in order to evoke an effective tumor antigen-specific immune response. Here, we found that the expression of Gal-9 in DCs was significantly knocked down after the "FSGG/siGal-9+M" treatment, as compared to the "FSGG/siGL2+M"

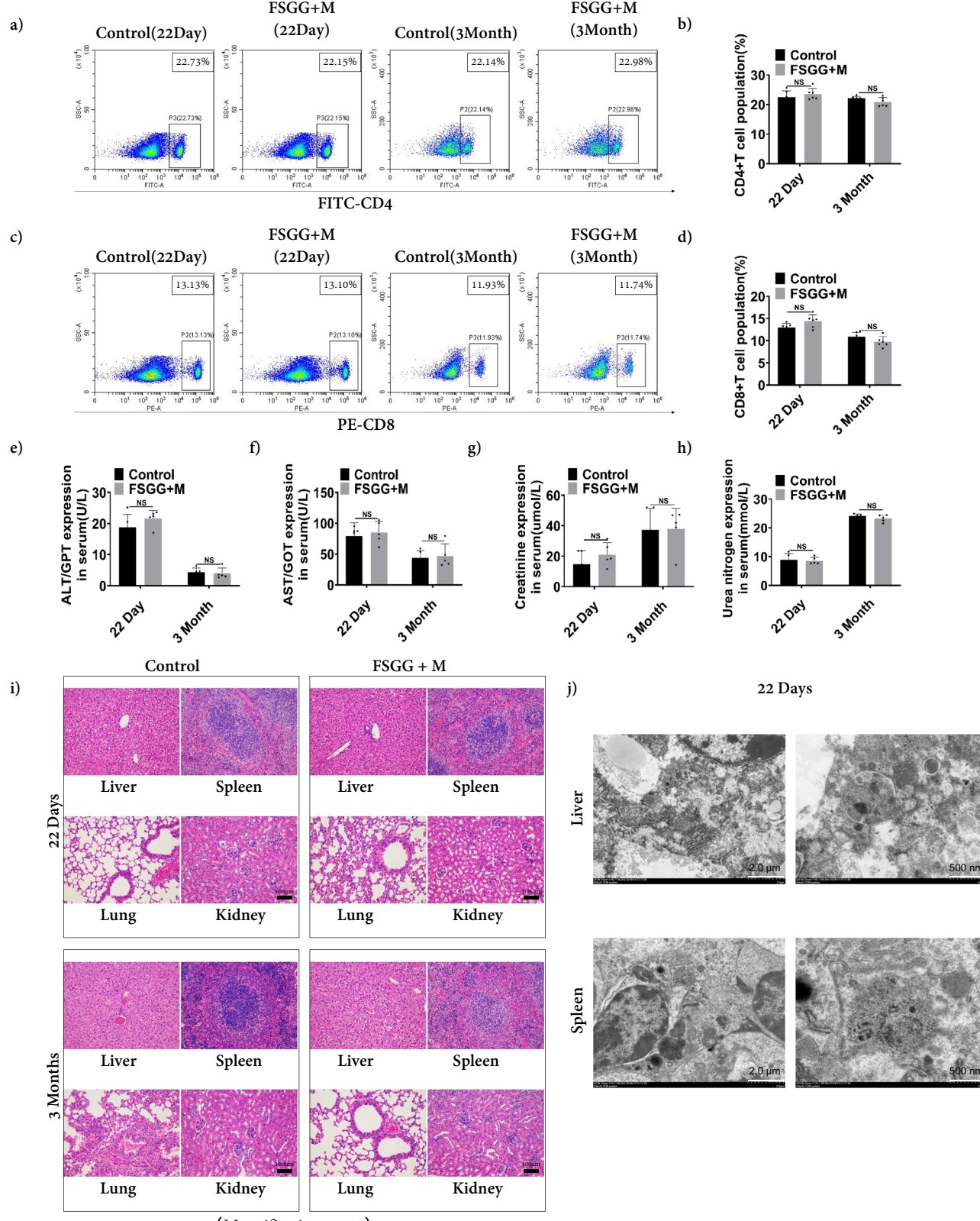

**Fig. 4 | Assessment of immunotoxicity and toxicity of FSGG in vivo.** The right back flanks of female C57BL/6 mice were topically treated with mixtures (Containing 25 μl Glycerol, 5 μl DMSO, 20 μl FSGG (6 μg/μl) or PBS (Control)) plus magnet field on every 3 days for 22 days, and then the mice were sacrificed after the treatment period or prolonging to 3 months from the starting. **a–d** Splenic CD4[+] and CD8[+] T cell percentages. Percentages of CD4[+] and CD8[+] T cells in spleens were analyzed by flow cytometry. The correspondent percentages of CD4[+] T cells (**b**) and CD8[+] T cells (**d**) were calculated. AST/GPT, AST/GOT, Creactinine, and Urea nitrogen tests. The total blood was collected, and the AST/GPT (**e**), AST/GOT (**f**), Creactinine (**g**), and Urea nitrogen (**h**) in serum were analyzed using the assay kits. **i** Histological images of the liver, spleen, lung and kidney were taken followed by paraffin section and H&E stain. **j** TEM images of liver and spleen from the 22-day FSGG-treating mice. Error bars represent the standard deviation of 3 experiments (NS, P > 0.05; Adding the magnet field is denoted with "+M").

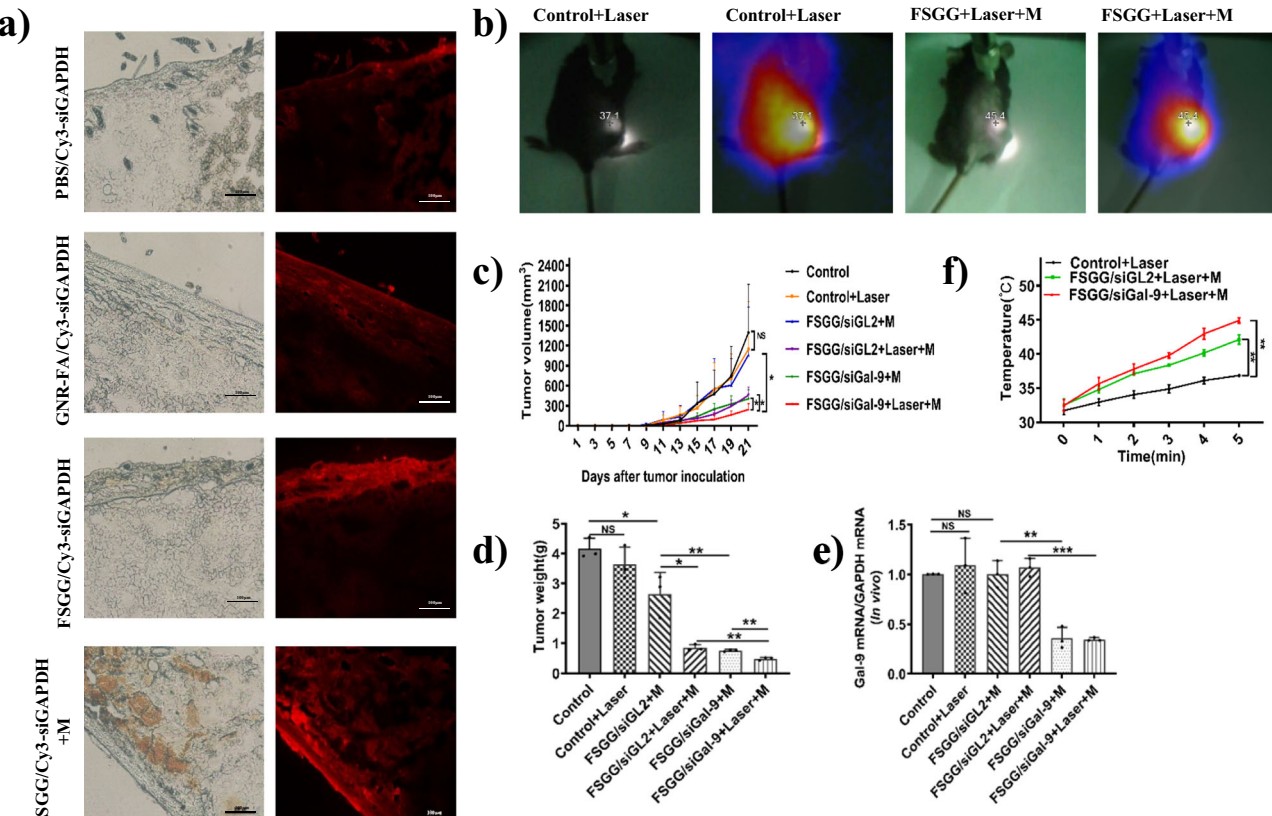

**Fig. 5 | In vivo topical siRNA delivery, PTT effects and photothermal-immunotherapy using FSGG/siGal-9 for melanoma.** B16-F10 melanoma tumor cells were inoculated into C57BL/6 mice as described in Materials and Methods. After 1 week when tumors are established, mice were anesthetized and the FSGG/siRNA mixtures (Glycerol: DMSO: FSGG/siRNA at 5:1:4 volume ratio) were topically applied on the top of tumors in the presence of magnet field. **a** In vivo topical tumor-specific delivery of siRNA by FSGG. For observing the topical siRNA delivery of FSGG/siRNA, we applied tumors with Cy3-labeled GAPDH siRNA followed by fluorescence microscope observation. Control mice applied with PBS or GNR-FA to substitute FSGG. 4 h after the topical application, the mice were sacrificed and the tumors were collected. The tumors tissues were frozen sectioned and visualized with a fluorescence microscope. **b** In vivo photothermal effects of FSGG. The tumor-bearing mice were topically treated with FSGG and PBS for 24 h. The treated sites of the mice were irradiated with 808 nm NIR laser at 1 W/cm² for 5 min, and the temperature was recorded with an infrared thermometer. Here, we showed the temperatures during the 5 min. **c–f** In vivo synergized photothermal-immunotherapy of FSGG/siGal-9 for melanoma. We used FSGG and siGal-9 mixture, following the above preparation, to treat the B16-F10 melanoma-bearing mice on days 7, 10, 13, 16, and 19. Twenty-four hrs after each application, laser irradiation (1 W/cm², 5 min) was performed. The tumor sizes were measured with a caliper and tumor volumes were estimated using the formula: tumor volume = (length × width²)/2.Tumor growth curves were plotted (**c**), and the tumor weights were measured (**d**). The gene silencing efficiencies in tumors were determined using RT-q-PCR (**e**). The temperatures induced by photothermal effects of FSGG/siRNA in vivo were measured using an infrared thermometer for 300 s and temperatures were plotted (**f**). Error bars represent the standard deviation of 3 experiments (*$P \leq 0.05$; **$P \leq 0.01$; ***$P \leq 0.001$; Adding the magnet field is denoted with "M").

treatment (8.58 ± 1.73% v.s. 14.91 ± 3.35%) (Fig. 6a, b), although the percentages of overall DCs were no significant changes in all the treated mice (Supplementary Fig. 9a, b). We in surprise observed that the "FSGG/siGL2+Laser+M" treatment significantly suppressed the DC Gal-9 expression compared to the "FSGG/siGL2+M" treatment (7.84% v.s. 12.18%; Fig. 6a, b), due to the photothermal effects. Besides, the combination treatment "FSGG/siGal-9+Laser+M" further significantly suppressed the level of Gal-9 in DCs compared to the single treatment "FSGG/siGal-9+Laser+M" or "FSGG/siGal-9+M", due to synergies of photothermal effects and knockdown of Gal-9, possessing the greatest suppression capacity by decreasing the percentage to 4.60% (Fig. 6a, b).

To determine whether FSGG/siGal-9 could rescue T cells from apoptosis, we collected splenic T cells from the B16-F10 cancer-bearing mice after photothermal-immunotherapy using FSGG/siGal-9. As shown in Fig. 6c, d, the percentages of apoptotic T cells were significantly decreased after the "FSGG/siGal-9+M" treatment compared to the "FSGG/siGL2+M" treatment (9.68% v.s. 21.37%). Notably, the combination treatment "FSGG/siGal-9+Laser+M" further significantly suppressed the T cell apoptosis (7.60%), compared to the individual treatment of photothermal effects "FSGG/siGL2+Laser+M" or silencing Gal-9 "FSGG/siGal-9+M" as well (Fig. 6c, d).

Gal-9 not only induces T cell apoptosis but also alters macrophages to an anti-inflammatory state and downregulates the number of effector T cells. Here, we analyzed the level of anti-tumor inflammatory cytokines such as IL-2, TNF-α and IFN-γ in the B16-F10 cancer-bearing mice after the treatment with photothermal-immunotherapy by FSGG/siGal-9. As shown in Fig. 6e–g, IL-2, TNF-α and IFN-γ in spleen were significantly increased by 4.37-fold, 2.57-fold, and 2.46-fold after the "FSGG/siGal-9+M" treatment compared to the "FSGG/siGL2+M" treatment due to silencing Gal-9; but the "FSGG/siGL2+Laser+M" treatment just increased the secretion of IL-2 by 2.39-fold while did not affect the expression of TNF-α and IFN-γ compared to the "FSGG/siGL2+M" treatment due to PTT, in addition, there is no significant difference of inflammatory cytokine release between the "FSGG/siGal-9+Laser+M" treatment and the "FSGG/siGal-9+M" treatment as well. Nevertheless, the mice after the "FSGG/siGL2+Laser+M" treatment similarly showed the highest levels of the cytokines (IL-2, TNF-α and IFN-γ) in splenic cell (Fig. 6e–g) or sera (Supplementary Fig. 9c–e).

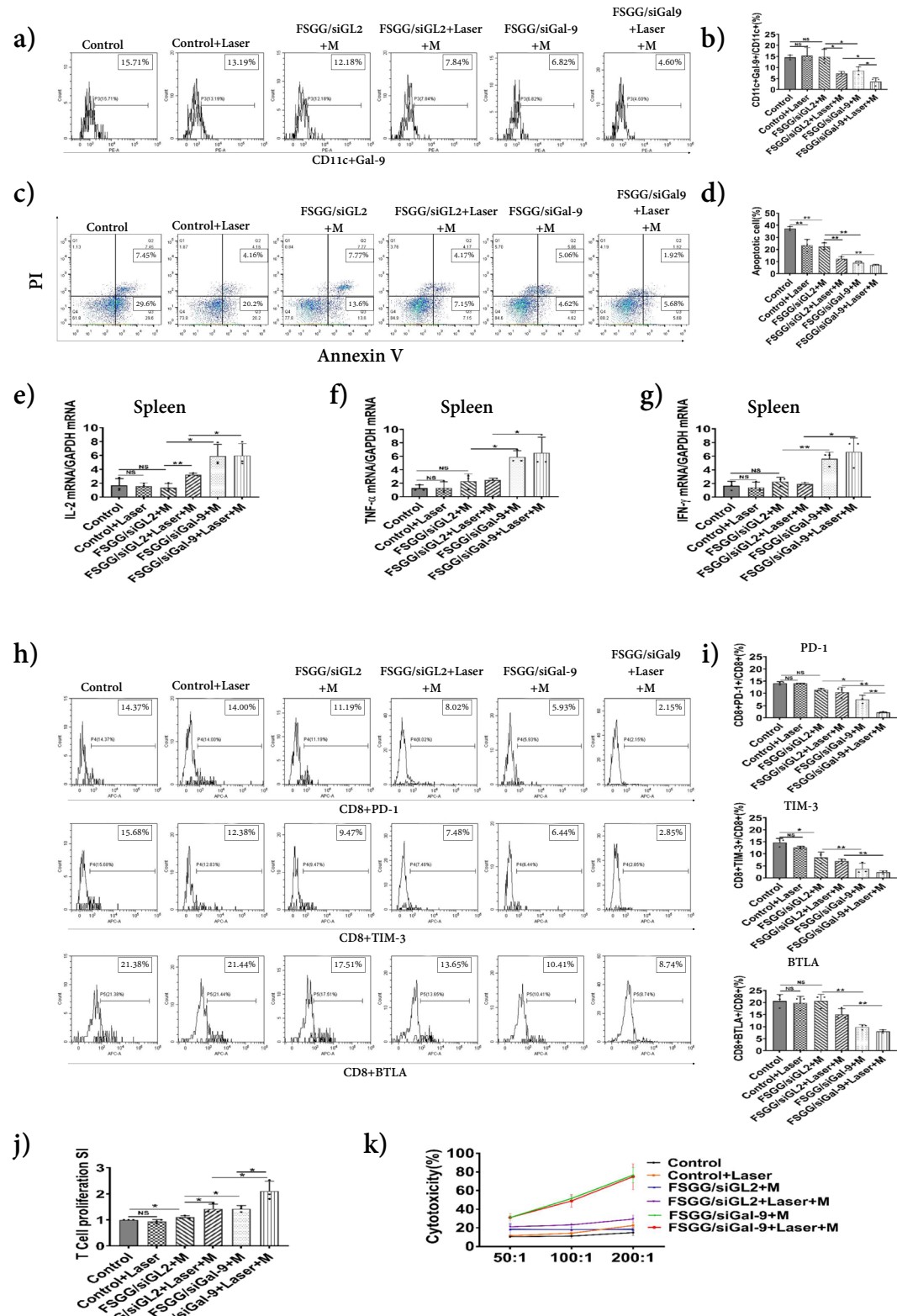

Expression of inhibitory receptors (IRs) PD-1, TIM-3 (HAVCR2), and BTLA was significantly higher on CD8[+] T cells isolated from tumor microenvironment, which is the major feature of exhaustive T cells. In order to determine whether the silencing of the Gal-9 gene is capable of reversing T cell exhaustion, the expression of inhibitory receptors on splenic CD8[+] T lymphocytes in of the tumor-bearing mice were detected by flow cytometry.

The results (Fig. 6h, i) showed that photothermal effects ("FSGG/siGL2+Laser+M" v.s. "FSGG/siGL2+M") had no significantly decreased the levels of IRs; while knocking down Gal-9 ("FSGG/siGal-9+M" v.s. "FSGG/siGL2+M") greatly decreased IRs levels; but the combination treatment "FSGG/siGal-9+Laser+M" exhibited the most significant suppression on the IRs expression (PD-1, 2.15%; TIM-3, 2.85%; BTLA, 8.74%) perhaps due

**Fig. 6 | Enhancement of anti-tumor immunity by FSGG/siGal-9 photothermal-immunotherapy.** Melanoma-bearing mice were treated as described in Fig. 5c–f. Mouse splenic cells and sera were collected as described in Materials and Methods. **a, b** Suppression of Gal-9 expression in splenic DCs. DCs in spleen were isolated and Gal-9 expression was detected by flow cytometry after CD11c-FITC and Gal-9-PE staining. The percentages of DC population and Gal-9 expression were calculated. **c, d** Suppression of splenic T cell apoptosis. Apoptosis of splenic cells were analyzed by flow cytometry after PI and Annexin V staining. The correspondent apoptotic percentages of T cells were calculated and results represent one of three experiments. Upregulation of inflammatory cytokines in spleen. The levels of IL-2, TNF-α, and IFN-γ of splenic cells were analyzed by RT-q-PCR (**e**), (**f**) and (**g**). **h, i** Reversing CD8$^+$ T cell exhaustion. The expression levels of PD-1, TIM-3, and BTLA on CD8$^+$ T cells were analyzed by flow cytometry. **j** Enhancing tumor-Ag-specific T cell proliferation. Splenic T cells were incubated with tumor antigen from B16-F10 cell lysates for 72 h. T cell proliferation was analyzed by CCK8 assay kit. **k** Enhancing CD8$^+$ T cell tumor-specific cytotoxic lysis. CD8$^+$ T cells from tumor mice were isolated using immunologic magnetic beads, and then incubated with B16-F10 cells as described in Materials and Methods. After 4 h of the incubation, supernatants were harvested and reacted with LDH reagents. The absorbance value was measured at a wavelength of 450 nm, and the cytotoxic killing of tumor cells was calculated. Error bars represent the standard deviation of three experiments (n.s. $P > 0.05$; *$P \leq 0.05$; **$P \leq 0.01$; Adding the magnet field is denoted as "M").

to the synergistic effects, which was also corroborated by RT-q-PCR test in spleen (Supplementary Fig. 9f–h). All these data indicated that the combinational FSGG/siGal-9 photothermal-immunotherapy could effectively reverse T cell exhaustion.

Next, we investigated the capacity of FSGG/siGal-9 photothermal-immunotherapy in stimulating tumor-Ag-specific T cell response. It was discovered that exhausted T cells were responsive to reinvigoration by blockade of the co-inhibitory such as PD-L1/PD-1 signal. Here, we detect the effects of FSGG/siGal-9 photothermal-immunotherapy on tumor-Ag-specific T cell response. The results (Fig. 6j) showed that compared with the "FSGG/siGL2+M" treatment, tumor-Ag-specific T cell response was increased only 29.41% by the "FSGG/siGL2+Laser+M" treatment due to photothermal effects, and only 29.59% by the "FSGG/siGal-9+M" treatment after knocking down Gal-9, but 91.23% by the "FSGG/siGal-9+Laser+M" treatment which further verified the synergistic FSGG/siGal-9 photothermal-immunotherapy.

Finally, we also assessed the tumor-specific cytotoxicity of T cells after FSGG/siGal-9 treatments. Firstly, the CD8$^+$ T cells (CTL) were isolated from the above treated mice and then co-cultured with B16-F10 melanoma cells ex vivo. In contrast with the "FSGG/siGL2+M" treatment (Cytotoxicity, 18.36 ± 1.70%), the photothermal effects of "FSGG/siGL2+Laser+M" treatment (Cytotoxicity, 29.42 ± 4.42%) didnot increased the cytotoxicity significantly; but the cytotoxicities of the FSGG/siGal-9 groups ("FSGG/siGal-9+M" and "FSGG/siGal-9+Laser+M") vigorously increased to 76.64 ± 8.34% and 74.96 ± 13.69%, compared to 14.85 ± 3.20% and 22.58 ± 3.06% of the PBS control groups ("Control" and "Control+Laser"), respectively (Fig. 6k). Taken together, these data suggest that photothermal-immunotherapy with FSGG/siGal-9 convincingly promoted the tumor-Ag-specific cytotoxicity of T cells.

## Discussion

Photothermal therapy (PTT) is a non-invasive anti-cancer method that exposes biological tissue to a temperature of 41–47 °C to promote the destruction of abnormal cells by using photothermal agents to convert photon energy into cytotoxic heat for cancer treatment without damaging healthy tissues. Compared with ablation methods, PTT can achieve a highly cellular degree of localization and applied to cancer areas where surgery is difficult. The efficacy of photothermal therapy depends on the energy absorption of laser irradiation and the heat conversion efficiency of nano-materials. Near-infrared (NIR) laser irradiation is considered to be the most suitable laser irradiation method in photothermal therapy, which can penetrate deep tissues to the maximum. In order to improve the efficiency and selectivity of energy to heat conduction, it is necessary to introduce a light-absorbing material (a photothermal agent) into the tumor. Gold nanorods have the longitudinal surface plasmon resonance (LSPR), which are often used as near-infrared laser excited nanomaterials, but the synthesis of gold nanorods is necessary CTAB has obvious cytotoxicity. In order to reduce the cytotoxicity of the material, we used MUA to modify the surface of GNR. The connection between CTAB and GNR was replaced by the more binding AU-S bond, so as to reduce the biological toxicity caused by CTAB on the surface of GNR. Besides, iron oxide magnetic nanoparticles have been widely used in biomedical applications such as magnetic resonance imaging, image-guiding cancer therapy, magnetically triggered drug release, which

were used as the core of FSGG nanostructure for topically transdermal delivery via the magnet force in our work. Mesoporous silica nanoparticles (MSNs) possessing high surface areas and tunable porosity parameters have attracted worldwide attention for the diagnosis and therapy of various diseases. MSNs have received FDA approval for first-in-human trials as a drug for cancers, which was synthesized as the biocompatible shell on the iron oxide nanoparticle cores to provide high surface areas for drug loading. Glucose transporters (GLUTs) has been reported in numerous cancer types including melanoma. Thus, we modified FS (Fe$_3$O$_4$@SiO$_2$) with PEI-PEG-glucose for cellular uptake, which can carry such the large biomolecules as siRNA and proteins due to highly positive-charged PEI and large surface area of mesoporous silica shell. Small interfering RNA (siRNA), a type of RNA molecule with a length of 20-25 nucleotides, effectively regulates gene expression through RNA interference (RNAi) to specifically silence the target messenger RNA (mRNA), so as to achieve the purpose of gene therapy. In this study, we successfully prepared a promising topically transdermal nanodrug "Fe$_3$O$_4$@SiO$_2$@PEG-PEI-glucose@GNR/Gal-9 siRNA" (FSGG/siGal-9, Fig. 2) for cancer therapy (IR = 88.24%, Inhibitory Ratio = (Average tumor weight of control group- Average tumor weight of experimental group)/Average tumor weight of control group ×100%; Fig. 5d), possessing the above mentioned multiple desirable and favorable priorities for in vivo application and anti-cancer therapy, namely composition of photothermal effect module gold nanorods (Figs. 2f, j, k, 5b, f), iron oxide for magnet-guiding topical transdermis (Fig. 5a), and glucose-based cellular targeting (Fig. 2g, h), and the gene silencing factor siGal-9 (Fig. 3a, c). In addition to these structural advantages, FSGG/siGal-9 also exhibited with good in vivo biocompatibility (Fig. 4e–i) and no immunotoxicity (Fig. 4a–d) and no accumulation in the body (Fig. 4j).

Metastatic melanoma, a kind of malignant tumor, is the most fatal skin cancer with the 80% mortality rate. The clinical treatment of melanoma mainly relies on traditional surgery, radiotherapy and chemotherapy, but these methods have unsatisfactory results, and side effects and poor prognosis. Therefore, finding effective treatments for melanoma is highly clinically demanding. Recent studies have shown that the occurrence and development of tumors are closely related to the immune function of the body, especially T lymphocytes in the tumor microenvironment (TME). As an important part of cellular immunity, T lymphocytes are a key factor in whether the body can successfully resist tumor invasion. At the same time, it also regulates with humoral immunity to kill tumor cells and inhibit tumor growth. After stimulated by antigen-presenting cells, initial T cells proliferate and exert effector functions, including releasing effector cytokines and cytotoxic lysis to kill tumor cells through Fas/Fas-L pathway. However, exhaustive T cells are a state of T cell function defects, often occurring in tumors and chronic infections, mainly manifested by the high expression of such IRs as PD-1, TIM-3, BTLA, CTLA-4, LAG-3, and TIGIT, and the gradual decline in the secretion of functional cytokines, including IL-2, IFN-γ, TNF-α, which can cause T cell dysfunction, induce T cell inactivation, and weaken the killing ability of T cells to tumor cells in the tumor micro-environment, leading to the occurrence of tumor immune escape. For the cancer immunotherapy, it is mainly by blocking the immune checkpoints to reverse the immune response ability of exhaustive T cells, and then improving the effect of anti-tumor treatment. For example, two anti- PD-1 antibodies, nivolumab and pembrolizumab, were approved by the US FDA

in 2014 for the treatment of metastatic melanoma but only achieved 30–40% objective response in patients. In 2019, studies discovered that Gal-9/TIM-3 pathway is the key mechanism of resistance to anti-PD-1 immunotherapy in lung cancer patients. Gal-9 is a type of carbohydrate-binding protein, as a member of the galactose agglutination family, and has β-galactose binding activity. Gal-9, a ligand of TIM-3 as a type of T cell surface inhibitory molecule specifically expressed on Th1 cells, mainly binds to TIM-3 positive cells to play suppressive roles in anti-tumor immunity, such as inhibiting the amplification of Th1 and Th17, promoting the apoptosis of Th1 cells and inducing the exhaustion of $CD8^+$ T cells. Thus, the interaction of Gal-9/TIM-3 directly or indirectly promotes the peripheral immune tolerance, weaken the body's anti-tumor immune response in the tumor microenvironment, thus leading to tumor immune escape. Blocking the signal transduction of the Gal-9/TIM-3 pathway can be used as a therapeutic target for clinical applications. This study firstly explored to bioinformatic analysis of Gal-9 in human malignant melanoma (Fig. 1). The TCGA data showed that expression level of Gal-9 was much higher expression levels than other immunosuppressive molecules such as CD80/86 and CD274 (PD-L1) in primary soild tumors (Fig. 1d and Supplementary Fig. 3). The single cell data showed Gal-9 (*LGALS9*) was higher expressed on the melanoma cells and immune cells (such as cytotoxic T cells, exhausted T cells, Treg, T memory cells, B cells, NK cells and monocytes) (Fig. 1c), and its receptor TIM-3 (*HAVCR2*) was relatively much higher expressed on the cytotoxic T cells, NK cells and exhausted T cells (Fig. 1c), moreover the cell communication in TME (Fig. 1b) exhibited that Gal-9/TIM-3 may be the key reason for cytotoxic T cells exhausting as well, thus leading to immune escape of acral melanoma. In the study, we blocked the expression of Gal-9 on tumor cells by RNAi gene silencing technology via our novel integrated nanostructure FSGG/siGal-9 for promoting anti-tumor immunity. Our results demonstrated that FSGG/siGal-9 nanostructure effectively knocked down the immune checkpoint Gal-9 in vitro (Fig. 3a, c) and in vivo (Fig. 5e). Besides, our results showed that the FSGG/siGal-9 application also suppressed the immune checkpoint Gal-9 in splenic DCs (Fig. 6b). Knocking down Gal-9 apparently reduced T cell apoptosis (Fig. 6d), increased inflammatory cytokine release of IL-2 (2.51 folds), IFN-γ (2.35 folds), TNF-α (3.64 folds) in spleen (Fig. 6e–g), reversed $CD8^+$ T cell exhaustion (Fig. 6i), and enhanced tumor-Ag-specific T cell response by 42.65% (Fig. 6j) and $CD8^+$ T cell tumor-specific cytotoxic lysis (4.16 folds) (Fig. 6k), and finally vigorously inhibited tumor growth in vivo (Fig. 5c).

Recently, PTT is used for not only primary local tumor (IR = 79.41% in our study; Fig. 5c), but also for distal metastatic tumors such as occult or scattered lesions via "abscopal effects". Unfortunately, PTT-treatment also elevated the expression of the immune checkpoints such as PD-L1 level of tumor cells in vitro, which impairs anti-tumor immunity, resulting in the impediment of PTT therapy efficacy[37]. There is a dilemma about the anti-tumor immunity induced by PTT. Our results firstly discovered that PTT increased anti-tumor immunity on the basis of effectively suppressing Gal-9 in splenic DCs (Fig. 6b), reducing T cell apoptosis (Fig. 6d), inducing IL-2 release in spleen (Fig. 6e), enhancing tumor-Ag-specific T cell response (Fig. 6j). Although PTT was significantly inhibited Gal-9 in splenic DCs but no decrease in melanoma cell in TME (Fig. 5e). Besides, some reported that heat (43 °C)-treated tumor cells were induced to release more exosomes. We also observed that thermal condition (44 °C) induced melanoma cells releasing more exosomes (Fig. 3d), and those tumor-derived exosomes can evoke a potent tumor antigen-specific immune response in vitro (Fig. 3e) perhaps due to carrying such as the tumor-specific antigen-MHC complex[31]. Besides, tumor-derived exosomes also carries such immune checkpoint as PD-L1[30]. Our studies presented that exosome possesses higher stimulating capacity for tumor-specific T cell proliferation than Ag, while Gal-9-silenced exosome (exosome/siGal-9) further enhanced this stimulating capacity (Fig. 3f). The proteomic analysis (Fig. 3g) illustrated that exosome/siGal-9 had a changed protein pattern that was related with such biological functions as viral carcinogenesis, human cytomegalovirus infection, carbon metabolism, Fc gamma R-mediated phagocytosis, leukocyte transendothelial migration, thus leading to the enhancement of the stimulating capacity.

Photothermal-immunotherapy (PTIT) means inhibiting immune checkpoints in tumor cells and tumor-derived exosome for producing synergistic effects on PTT. The combination treatment of PTT and silencing Gal-9 with FSGG/siGal-9 plus laser irradiation plus magnet field ("FSGG/siGal-9+Laser+M") further suppressed tumor growth (Fig. 5c) and decreased the tumor weight (Fig. 5d), compared to the mice receiving single treatment with the photothermal effects ("FSGG/siGL2+Laser+M") or gene silencing of Gal-9 ("FSGG/siGal-9+M"). The ex vivo anti-tumor immunity exhibited that the combination treatment also significantly suppressed Gal-9 in splenic DCs (Fig. 6b), reduced PD-1 on the $CD8^+$ T lymphocytes (Fig. 6i), and increased tumor-Ag-specific T cell response (Fig. 6j), thus leading to PTIT.

Conclusively in this study, we successfully constructed a novel integrated nanodrug FSGG/siGal-9 ($Fe_3O_4$@$SiO_2$@PEG-PEI-glucose@GNR/Gal-9 siRNA) for transdermally treating melanoma via achieving PTIT. We firstly validated that Gal-9/TIM-3 is a new effective target for melanoma immunotherapy perhaps because of being high expression and inhibiting cytotoxic T cells in TME. PTT increases anti-tumor immunity in vivo, and PTIT achieves synergistic effects on melanoma treatment.

## Materials and methods
### Cell lines and mice
A murine melanoma cell line B16-F10 was obtained from the American Type Culture Collection (ATCC). Cells were maintained in DMEM medium (Life Technologies, Carlsbad, CA) containing 10% FBS at 37 °C in 5% $CO_2$. Female C57BL/6 mice (6 to 8 weeks old) were purchased from Hunan Lake Jingda Laboratory Animal Co., Ltd. and were kept in a SPF (Specific pathogen free) grade animal center under the condition of 23 ± 2 °C and 50 ± 5% humidity. The use of the mice complied with the Regulations for the Administration of Affairs Concerning Experimental Animals of China. All animal experiments received the ethical approval of Animal Care and Use Committee of Nanchang University, China. We have complied with all relevant ethical regulations for animal use.

### Preparation of Gal-9 siRNA
The sequence of siRNA targeting Gal-9 gene is 5′-CTGAA-GAACTTGCAGGATA-3′ and was synthesized by Dharmacon, Inc. (Lafayette, CO, USA). The commercially available targeted luciferase gene siGL2 was used as a negative control siRNA.

### Preparation of $Fe_3O_4$@$SiO_2$ (FS)
The preparation method of $Fe_3O_4$ modified with oleic acid was as follows: 8.1 g $FeCl_3 \cdot 6H_2O$ (Shandong West Asia Chemical Co., Ltd., Shandong, China) was added into 142.5 ml deionized water, and heat it to 70 °C while stirring until being completely dissolved. Then 4.4 g $FeCl_2 \cdot 4H_2O$ (Shandong West Asia Chemical Co., Ltd.) was dissolved into 10 ml deionized water and filtered, and transfer 7.5 ml filtrate to the above $FeCl_3$ solution. Then, 18 ml of 25% ammonia water (Shandong West Asia Chemical Co., Ltd.) was rapidly added while stirring vigorously. After 1 min, 4.66 g of oleic acid (Shandong West Asia Chemical Co., Ltd.) was dropped, and the black colloidal substance was obtained by rapid stirring at 70 °C for 1 h. Under the condition of magnetic field, the precipitate was collected and cleaned repeatedly with ethanol (Shandong West Asia Chemical Co., Ltd.) and deionized water. After freeze-drying for 40 h, the oleic acid modified iron oxide powder was obtained.

The preparation of $Fe_3O_4$@$SiO_2$ was conducted by following procedures: 0.292 g CTAB (Sinopharm Shanghai Chemical Reagent Co., Ltd., Shanghai, China) was dissolved in 10 ml deionized water. Then 0.0510 g $Fe_3O_4$ powder was dispersed in 1.2 ml chloroform (Shandong West Asia Chemical Co., Ltd.) by ultrasonic for 5 min, and then slowly dropped into CTAB solution, which was ultrasonicated for 40 min (50 °C, 100 W) until the color became bright black. The black solution was transferred into a three necked flask, then adding 45 ml deionized water and 0.6 ml, 1 M NaOH solution (Shandong West Asia Chemical Co., Ltd.), closed stirring at 70 °C. After 10 min, 0.5 ml TEOS (Shanghai Aladdin Biochemical

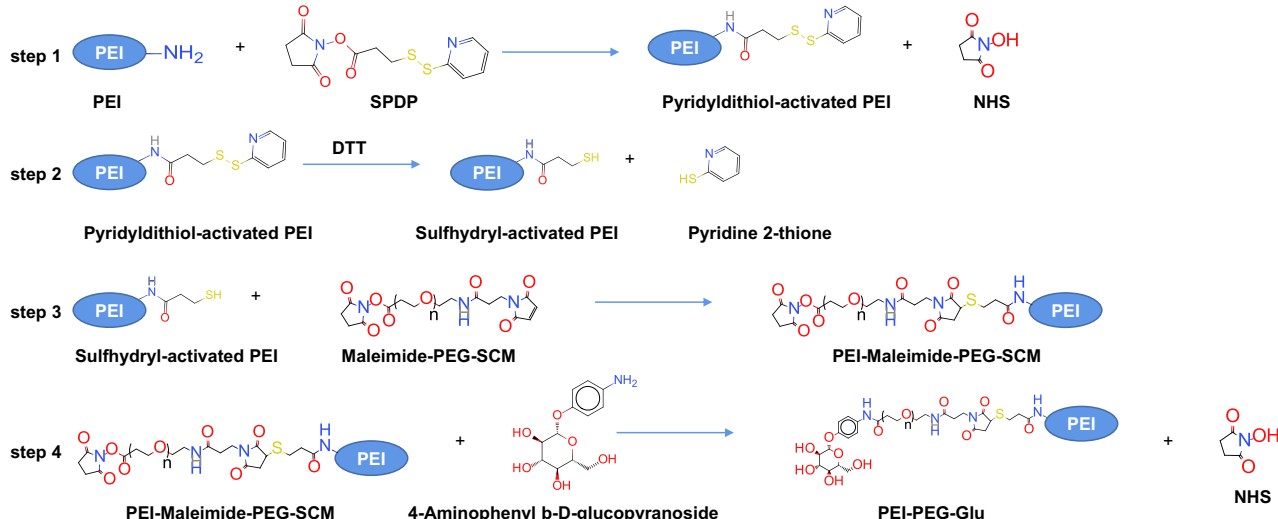

**Fig. 7 | The reaction steps of are listed PEI-PEG-Glu.** PEI are modified with SPDP reagent (step 1), and then the reaction products is treated with reducing agent DTT to expose sulfhydryl groups (step 2), and finally the Sulfhydryl-activated PEI are conjugated with PEG ($n = 24$) (step 3) and glucopyranoside to achieve PEI-PEG-Glu (step 4).

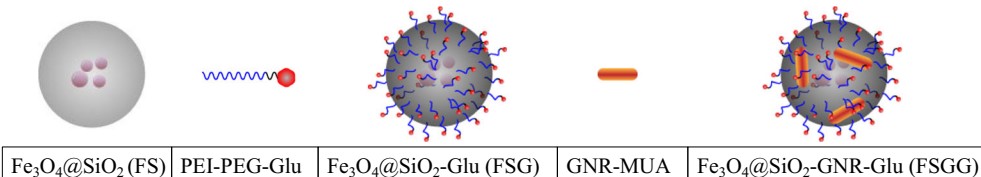

| Fe₃O₄@SiO₂ (FS) | PEI-PEG-Glu | Fe₃O₄@SiO₂-Glu (FSG) | GNR-MUA | Fe₃O₄@SiO₂-GNR-Glu (FSGG) |

**Fig. 8 | The preparation process of Fe₃O₄@SiO₂-GNR-Glu (FSGG) by the lay-by-lay method (LBL).** Firstly, the prepared Fe₃O₄@SiO₂ (FS) and PEI-PEG-Glu were bound to Fe₃O₄@SiO₂-Glu (FSG). Then, FSG bound GNR-MUA to form the final nanostructure Fe₃O₄@SiO₂-GNR-Glu (FSGG). The whole processes were verified by Zeta potential and UV-Vis absorption spectra.

Technology Co., Ltd., Shanghai, China) was slowly added into stirring for 10 min, and then 3 ml ethyl acetate (Shanghai McLin Biochemical Technology Co., Ltd., Shanghai, China) was slowly added into stirring for 3 h more. Finally, this mixture was cooled to room temperature, and the transparent yellow supernatant was taken out and frozen-dried to obtain Fe₃O₄@SiO₂ nanoparticle.

**Preparation of GNR-MUA**

The synthesis of water-soluble gold nanorods (GNR) is through a seed-mediated growth route. Initially, 1 ml, 0.2 M CTAB was mixed with 1 ml, 0.5 mM HAuCl₄ (Sinopharm), and then added 0.12 ml, 0.01 M of pre-cooled NaBH₄ (Sinopharm) while stirring at 28 °C, until the resulting solution is brown and stand still for 2 h before use. Subsequently, the preparation method of the growth solution is as follows: 50 ml, 0.2 M CTAB, 50 ml, 1 mM HAuCl₄ and 2.5 ml, 4 mM AgNO₃ (Sinopharma) were lightly mixed at 28 °C. After fully mixed, the solution was added 670 µl, 0.079 M ascorbic acid (Chinese Medicine Group Chemical Reagent Co., Ltd., Shanghai, China) under stirring at 28 °C until the color turn from dark yellow to colorless, and then added 120 µl seed solution into the growth solution. Then, the solution gradually changes from colorless to purple-red. After stirring at 28 °C for 24 h, the solution was centrifuged at 12,000 rpm for 10 min in order to remove additional CTAB, and then freeze-dried into GNR powder for storage.

Then, we prepared GNR-MUA using following methods: Mercaptoundecanoic acid (MUA; 440 mg) (Chinese Medicine Group Chemical Reagent Co., Ltd.) was added to 80 ml deionized water, and ultrasonically dispersed until the solution became a white suspension, and then added 1.6 ml of 20% NaOH, stirred slowly for 5 min and then became a transparent solution, and finally made up to 100 ml with water to obtain 23 mM MUA solution. Freeze-dried CTAB-GNR powder (100 mg) was suspended in 100 ml MUA solution and reacted for 24 h at room temperature. CTAB was successfully replaced MUA due to the Au-S bond. The solution was centrifuged at 12,000 rpm for 15 min, and vacuum freeze-dried into GNR-MUA powder for storage.

**Preparation of PEI-PEG-Glu**

200 mg PEI (MW: 250 KD) (Beijing Solaibao Company, Beijing, China) was dissolved into 20 ml methanol, and then added 50 µl SPDP (1 mg/ml) (Thermo Scientific, USA) dissolved in DMSO stirring at room temperature for 24 h, and then dialyzed (cutoff: 15 KD) (Beijing Solaibao Company, Beijing, China) in 600 ml deionized water for 24 h at room under the protection of DTT (60 mg) (Sigma aldrich, USA). Then, the dialysate was taken out and then added 0.52 ml, 0.5 mg/ml Maleimide-PEG-SCM DMSO solution (Sigma aldrich) and 43.4 µl, 0.05 mg/ml 4-Aminophenyl b-D-glucopyranoside DMSO solution (Sigma aldrich) in turns and then stirred for 24 h at room temperature (RT). Then the solution was dialyzed (Cutoff: 15 KD) in 600 ml deionized water for 24 h at room under the protection of DTT (60 mg). During the dialysis, the water was changed once. Finally, the complex was freeze-dried to obtain PEI-PEG-Glu powder. The reaction steps are listed in Fig. 7.

**Preparation of Fe₃O₄@SiO₂-GNR-Glu (FSGG)**

The FSGG complex was synthesized by the lay-by-lay method (LBL) as shown in Fig. 8. 30 mg Fe₃O₄@SiO₂ and 60 mg PEI-PEG-Glu were dissolved in 30 ml deionized water and stirred at 28 °C for 3 h to obtain Fe₃O₄@SiO₂-Glu (FSG). Under the magnetic field, the precipitate was collected and cleaned twice with deionized water to remove unbound PEI-PEG-Glu and then resuspended with 30 ml deionized water. Then 30 mg GNR-MUA was added into and stirred at 28 °C for 24 h to obtain Fe₃O₄@SiO₂-GNR-Glu (FSGG). Under the magnetic field, the precipitate was collected and cleaned

twice with deionized water to remove unbound GNR-MUA and then freeze-dried (Shanghai Yetuo Technology Co., Ltd, Shanghai, China) to obtain FSGG powder. Finally, the FSGG powder was resuspended into 1 mg/ml with deionized water for usage.

### Ultraviolet-Visible analysis
The GNR-MUA, $Fe_3O_4@SiO_2$ (FS), $Fe_3O_4@SiO_2$-Glu (FSG) or $Fe_3O_4@SiO_2$-GNR-Glu (FSGG) powder was dissolved in deionized water. The surface plasma resonance effect of nanomaterial in the wavelength range of 400 to 900 nm was observed by a multifunctional enzyme analyzer (Spectra Max M5e, MD, USA).

### Transmission electron microscopy (TEM)
The $Fe_3O_4@SiO_2$-Glu (FSG) or $Fe_3O_4@SiO_2$-GNR-Glu (FSGG) powder was dissolved in deionized water, and then was dropped on a copper grid and dried for 30 min. The morphology of the material was determined by transmission electron microscope (TEM) (LIBRA 120, Carl Zeiss, Germany).

### Zeta potential analysis
The GNR-MUA, $Fe_3O_4@SiO_2$ (FS), $Fe_3O_4@SiO_2$-Glu (FSG) or $Fe_3O_4@SiO_2$-GNR-Glu (FSGG) powder was dissolved in deionized water. The zeta potential of nanomaterial was measured by a Zetasizer Nano ZS (Malvern Instruments, Malvern, UK).

### Gel shift assays
Cy3-siGAPDH and FSGG were mixed according to the different mass ratio (1:0, 1:1, 1:2, 1:3, 1:4, 1:5, 1:6, 1:7). After incubating 30 min at room temperature, the samples were added to a 2% agarose gel containing 0.02% Goldview (Bioshop, USA) and electrophoresed at 80 V for 30 min. Then the encapsulation rate of Cy3-siGAPDH by FSGG was observed on gel imaging system.

### In vitro cytotoxicity in B16-F10 cancer cells
To determine the cell survival rate after incubation with FSGG, B16-F10 cells were used for detection by MTT method. B16-F10 cells ($5 \times 10^3$ /well) was seeded in 96-well plate and cultured at 37 °C, 5% $CO_2$ for 24 h. Then the different concentrations of 200 µl PBS as control or FSGG (10 µg/ml, 20 µg/ml, 30 µg/ml, 40 µg/ml, 50 µg/ml, 60 µg/ml, 70 µg/ml) in cell culture medium was used to culture cells for 24 h or 48 h. Then 20 µl MTT reagent was added to each well. After incubating for 4 h, culture medium was removed and 150 µl DMSO was added to each well. Then spectrophotometric analysis was performed at 490 nm using a reference of 650 nm with a microplate reader (Spectra-Max M5e, MD, USA).

### FSGG photothermal effects in vitro
1 ml of deionized water or GNR-MUA solution (40 µg/ml) in deionized water or FSGG solution (40 µg/ml) in deionized water was added to the 24-well plate, and then irradiated with 808 nm NIR laser at 2 W/cm². The temperature changes of solutions were monitored with an infrared thermometer every 30 s.

To determine the in vitro photothermal effects of FSGG. 200 µl B16-F10 cells ($5 \times 10^3$ /well) was seeded in 96-well plate and cultured at 37 °C, 5% $CO_2$ for 24 h. Then 200 µl PBS or FSGG (40 µg/ml) in cell culture medium was used to culture cells. After 24 h, the wells were irradiated by the 808 nm infrared laser at different optical densities (0 W/cm², 1 W/cm², 2 W/cm², 3 W/cm², 4 W/cm²) and irradiation time (0 s, 100 s, 200 s, 300 s, 400 s, 500 s). Then, the culture was continued for 24 h. 20 µl MTT reagent was added to each well. After incubating for 4 h, culture medium was removed and 150 µl DMSO was added to each well, then spectrophotometric analysis was performed at 490 nm using a reference of 650 nm with a microplate reader (Spectra-Max M5e, MD, USA).

### In vitro transfection of B16-F10 cells by FSGG/siRNA
B16-F10 cells ($1 \times 10^5$ /well) was seeded in 24-well plate and cultured at 37 °C, 5% $CO_2$ for 24 h. FSGG and Cy3-siGAPDH (mass ratio

(wt(FSGG):wt(Cy3-siGAPDH)) = 20:1) were mixed for 30 min and then diluted into 40 µg/ml of FSGG with opti-MEM medium. 500 µl the mixture (wt(Cy3-siGAPDH) = 1 µg, wt(FSGG) = 20 µg) was added into each well. After 4 h of transfection, the nuclei were stained with DAPI (Invitrogen, USA) staining agent, and the fluorescence intensity was observed under fluorescence microscope (Olympus, Model BX51, Japan). In addition, the cells were collected and the transfection efficiency was detected by flow cytometry (BD FACS Calibur, BD Biosciences, Mountain View, CA; Flow Cytometry Gating Strategy has been uploaded in the Supplementary Figs. 10–13.).

### TCGA (The Cancer Genome Atlas) database
Bulk rna_seq data in TCGA were downloaded through the Genomic Data Commons (GDC) Data Portal, RPM value and clinical data were extracted and applied.

### Single-cell sequencing analysis
Single cell rna_seq data of three acral melanoma patients were obtained from the GEO database under accession code GSE215120 (GSM6622299, GSM6622300, GSM6622301)[38]. The cells were clustered from the following parameters (nfeatures = 4000 for the FindVariableFeatures function, npcs = 100 for the RunPCA function (Supplementary Fig. 1a), dims = 70 for the FindNeighbors function, and resolution = 1.5 for the FindClusters function). The resulting single-cell clusters were visualized in t-SNE (dims = 70) representations and annotated to biological cell types by canonical marker genes (Supplementary Table 2)[39]. The immune checkpoint communications on the cells in the tumor microenvironment was plotted using the iTalk package[40].

### Real-time quantitative PCR (RT-q-PCR)
Trizol (Trizol reagent, Invitrogen) was used to extract total cellular RNA as a template for cDNA synthesis using reverse transcriptase (MMLV-RT, Invitrogen Life Technologies). RT-q-PCR was performed using gene specific primers in the stratagene Mx3000p RT-q-PCR system (Agilent Technologies, Lexington, MA, USA). The housekeeping gene GAPDH was used as internal reference. The primer sequences were as follows: GAPDH forward: 5′-GGCAAATTCAACGGCACAGT-3′ and reverse: 5′-GTCTCGCT CCTGGAAGATGG-3′; Gal-9 forward: 5′-GTTGTCCGAAACACTCAGA T-3′ and reverse: 5′-ATATGATCCACACCGAGAAG-3′; IL-2 forward: 5′-ACATTGACACTTGTGCTTGTGG-3′ and reverse: 5′-TTGAGGGCTT GTTGAGATGATGCT-3′; TNF-α forward: 5′-GCCTCTTCTCATT CCTGCTTGTGG-3′ and reverse: 5′-CCCGTTATCTCCCCTTCATCTT CC-3′; IFN-γ forward: 5′-GCTACACACTGCATCTTGGC-3′ and reverse: 5′-GCTTTCAATGACTGTGCCGT-3′; PD-1 forward: 5′-CCGCTTCCA-GATCATACAG-3′ and reverse: 5′-CTCTGGCCTCTGACATACTTG-3′; TIM-3 forward: 5′-AGTGGGAGTCTCTGCTGGGTTGA-3′ and reverse: 5′-AGGATGGCTGCTGGCTGTTGA-3′; BTLA forward: 5′-TGCAG-GAGCCAGAAGAGAAAGTCA-3′ and reverse: 5′-CAATGTGGGGGT-CAGGGATGG-3′. The RT-q-PCR was conducted using SYBR green PCR reagents (Invitrogen Life Technologies). The reactions of RT-q-PCR were amplified, in accordance to the manufacturer's protocol, in Stratagene Mx3000P QPCR System (Agilent Technologies, Santa Clara, CA, USA). The difference of gene expression was calculated by using the $2^{-\Delta\Delta Ct}$ method.

### Western blot
The total protein of B16-F10 cells was extracted by RIPA buffer (cell signaling, Pickering, Ontario, Canada). BCA Protein Assay kit (Beyotime, Shanghai, China) was used to detect protein concentration. The protein samples were separated by 10% SDS-PAGE gel and transferred to PVDF membranes. PVDF membrane was sealed with 5% BSA for 2 h and incubated with anti GAPDH or Galectin-9 antibody at 4 °C overnight. The PVDF membrane was washed three times with TBST and incubated with secondary antibody for 2 h at room temperature. The protein bands were detected with ECL Western blot substrate (Beyotime, Shanghai, China) with a ChemiDocTM MP Imaging System (Bio-Rad, Berkeley, CA, USA). Then

band intensities were measured with ImageJ software and normalized to GAPDH.

## In vivo immunotoxicity and toxicity

Transdermal drug delivery system (Volume ratio of Glycerol:DMSO:PBS or FSGG = 5:1:4; 50 μl mixture of 25 μl Glycerol, 5 μl DMSO, 20 μl FSGG (6 μg/μl) or PBS as control per mouse) plus magnet field was applied topically on the shaved right back flanks of female C57BL/6 mice (6 to 8 weeks old) every 3 days, and the health of the mice was regularly observed. The mice (10 mice per group) were sacrificed at 22 days and 3 months after topical application, and samples were collected for subsequent tests: (1) Percentage of CD4$^+$/CD8$^+$ T cells in spleen. T lymphocytes were isolated from spleen and incubated with anti-CD4 and anti-CD8 antibodies for 15 min in dark, and cells were sorted by flow cytometry using FACS Calibur (BD Biosciences, Mississauga, Ontario, Canada); (2) Toxicity assessment of liver and kidney tissues. After the collected eyeball blood was coagulated at room temperature and then centrifuged at 3000 rpm for 10 min, the supernatant serum was collected, and the expression levels of ATL/GPT, AST/GOT, CRE and BUN were determined according to the instructions of the kits (Nanjing Jiancheng Biotechnology Co., Ltd., Nanjing, China); (3) Histopathology Evaluation. The liver, spleen, lung and kidney were collected and fixed in paraformaldehyde (4% w/v in PBS). Then the tissues were embedded in paraffin blocks and cut into about 5 μm slices. After stained with hematoxylin and eosin (Sigma,St.Louis, MO), the histopathological changes of the sections were observed under the light microscope.

## Preparing exosome or exsosome/siGal-9 suspension

(1) 500 μl $2 \times 10^5$ cell/ml B16-F10 cell suspensions were seeded into 6-well plates per well. After 24 h, cells were transfected siGal-9 with lip2000, and then cultured in 5% $CO_2$ cell incubator at 37 °C for 4–6 h, and then the culture medium was replaced with 15% FBS DMEM. After 12 h, replacing the 15% FBS DMEM with a DMEM medium without FBS. Then incubating for another 12 h in a 37 °C or 44 °C cell incubator, and collect the cell supernatant;

(2) Centrifuging the cell supernatant at 4 °C 400 g for 10 min, removing the precipitation, and then syringely filtering with 0.22 μm aperture filter. Then, the cell supernatant was applied the Amicon ® Ultra (15 ml, 3 KD) filter, centrifuged at 4 °C 4000 g for 40 min, and finally concentrated into 1 ml from 120 ml;

(3) Then add 15 ml PBS in the Amicon ® Ultra filter, continue centrifugation, and finally obtain about 600 μl–1.5 ml exosome suspension stored at 4 °C for standby. Proteomics of exosome or exosome/siGal-9 (Supplementary Data) were analyzed by LC-MS/MS.

## Preparing tumor antigen (Ag)

Disperse $1 \times 10^6$ cells with 250 μl PBS, and obtain tumor antigen (Ag) suspension by repeatedly freezing and thawing the cells at −80 °C and 40 °C and then centrifuging the cell lysate at 4 °C 2000 g for 30 min.

## BCA method

(1) Preparation of BCA working solution: prepare the mixture according to the volume ratio of BCA working solution A to B of 50:1 for standby.

(2) Dilution standard: accurately measure 10 μl standard and dilute it with PBS to 0.5 mg/ml. Pipetting 0 μl, 1 μl, 2 μl, 4 μl, 8 μl, 12 μl, 16 μl, 20 μl of the standard in turn and add it to the corresponding well of the 96-well plate. Adding PBS to each well to make up to 20 μl. Then adding 200 μl BCA working solution to each well.

(3) Diluted sample (Ag, exosome, or exsosome/siGal-9): accurately measure 5 μl of sample per well to 96-well plate, similarly add PBS to make up to 20 μl, and add 200 μl BCA working solution to each sample wells.

(4) After shaking and mixing, the 96-well plate was placed in a 5% $CO_2$ incubator at 37 °C for 30 min.

(5) Determining the OD value of the standards and samples at 562 nm with a multi-function microplate reader, and calculating the protein concentration of each sample according to the drawn standard curve.

## Topical FSGG/siRNA delivery in vivo

The melanoma tumor B16-F10 cells ($7 \times 10^4$ cells in 100 μl) suspension were inoculated on the right back flanks of mice where had been shaved in advance. Transdermal drug delivery system (Volume ratio of Glycerol:DMSO:PBS/Cy3-siGAPDH or GNR-FA/Cy3-siGAPDH or FSGG/Cy3-siGAPDH = 5:1:4; 50 μl mixture of 25 μl Glycerol, 5 μl DMSO, 20 μl FSGG/Cy3-siGAPDH or GNR-FA/Cy3-siGAPDH or PBS/Cy3-siGAPDH as control per mouse, where wt(FSGG or GNR-FA):wt(siRNA) = 6:1 dissolving in PBS and wt(FSGG or GNR-FA) = 120 μg and wt(Cy3-siGAPDH) = 20 μg) is applied topically on the skins above inoculated tumors. Here, the synthesis of GNR-FA referred to our previous works. The magnet field was added in the group with the treatment of FSGG/Cy3-siGAPDH+M for 1 h. After 4 h, the mice were scarified and the tumors were collected. The tumors were quickly freeze-embedded using an optimal cutting temperature (OCT) compound (Triangle Biomedical Sciences, USA) on a freezing stage and cut into 5 μm horizontal sections. Images were visualized with a fluorescence microscope (Olympus).

## Treatment of B16-F10 melanoma cancer using photothermal-immunotherapy and photothermal effects in vivo

The right back flank of female C57BL/6 mice was shaved and the model of tumor bearing mice was established by inoculating 100 μl $7 \times 10^5$ /ml B16-F10 cells subcutaneously. When the diameter of tumor was ~3 mm after 1 week, the mice were randomized into six groups. The transdermal drug delivery system (Volume ratio of Glycerol:DMSO:PBS/siGal-9 or FSGG/siGal-9 = 5:1:4; 50 μl mixture of 25 μl Glycerol, 5 μl DMSO, 20 μl FSGG (6 μg/μl) or PBS as control per mouse) was applied topically to the shaved tumor site on the day 7, day 10, day 13, day 16 and day 19. After 24 h of each treatment, the mice were irradiated with 808 nm near-infrared laser (1 W/cm$^2$, 5 min), and the temperature increase induced by photothermal effects of GNR in vivo was measured every 1 min using an infrared thermometer for 300 s and temperatures were plotted. The tumor size was measured with digital vernier caliper every other day. The tumor sizes were measured with a caliper and tumor volumes were estimated using the formula: tumor volume = 1/2 (length × width$^2$). Tumor growth curves were plotted. On the 22 days, the mice were sacrificed and tumors, blood, and spleens were collected. The tumor weights were measured. The magnet field was added in the groups for 1 h after each application.

## T-cell apoptosis assays

100 μl of splenic lymphocyte suspension ($1 \times 10^7$ cells/ml) was resuspended in 200 μl 1× binding buffer, and incubated with 2 μl Annexin V and PI (eBioscience, San Diego,CA, USA) for 15 min at room temperature in the dark. Cells were sorted by flow cytometry using FACS Calibur (BD Biosciences, Mississauga, Ontario, Canada).

## Expression of Gal-9 protein in DC cells

100 μl of splenic lymphocyte suspension ($1 \times 10^7$ cells/ml) was resuspended in 100 μl PBS buffer, and stained with 0.5 μl of each of the following monoclonal antibodies: FITC-anti-CD11c, PE-anti-Gal-9 (Invitrogen, USA) at 4 °C for 30 min. After incubation, the cells were washed twice with PBS and resuspended with 300 μl PBS, and then detected by flow cytometry using FACS Calibur (BD Biosciences, Mississauga, Ontario, Canada).

## Expression of IRs in CD8$^+$ T lymphocytes

100 μl of splenic lymphocyte suspension ($1 \times 10^7$ cells/ml) was resuspended in 100 μl PBS buffer, and stained with 0.5 μl of each of the following monoclonal antibodies: PE-anti-CD8, APC-anti-PD-1, APC-anti-TIM- 3, APC-anti-BTLA (Invitrogen, USA) at 4 °C for 30 min. After incubation, the cells were washed twice with PBS and resuspended with 300 μl PBS, and then detected by flow cytometry using FACS Calibur (BD Biosciences, Mississauga, Ontario, Canada) .

## Ag, exosome, or exsosome/siGal-9 stimulating the proliferation of tumor Ag-specific lymphocytes

Splenic lymphocytes ($2.5 \times 10^3$ cells/well) were seeded in 96-well plate, and then 10 μg (BCA protein quantification, or 50 μg/ml) B16-F10 Ag, exosome, or exsosome/siGal-9 was added into each well. After culturing for 72 h, 10 μl CCK8 reagent (5 mg/ml) (BD Pharmingen, San Diego, CA, USA) was added to each well for another 4 h. The supernatant was moved into a new 96-well plate and the optical density (OD) was read at 450 nm with a microplate reader (Spectra-Max M5e, MD, USA).

## Tumor-specific cytotoxic lysis

The $CD8^+$ T lymphocytes were isolated using $CD8^+$ immuno-magnetic beads (Miltenyi Biotec, USA). B16-F10 cells ($5 \times 10^3$ /well) were co-cultured with $CD8^+$ T cells at the ratio of 1:50, 1:100 and 1:200 for 4 h (5% $CO_2$, 37 °C), and then the supernatt was harvested and mixed with LDH according to reagent Manufacturer's instruction of the CytoTox 96® non-radioactive cytotoxicity analysis kit (Promega, USA). 10 μl 10× Lysis Solution was added into $5 \times 10^3$ /well B16-F10 tumor cells as the maximum release references. The optical density in each well was measured at a wavelength of 490 nm with a microplate reader (Spectra-Max M5e, MD, USA). The percentage of specific cytotoxicity was calculated as follows: % Cytotoxicity = (Experimental OD value - $CD8^+$ T lymphocyte spontaneous OD value - B16-F10 cell spontaneous OD value)/ (B16-F10 cell maximum OD value - B16-F10 cell spontaneous OD value) ×100%.

## Statistical analysis

Data were presented as mean ± SD. Student's $t$-test (2-tailed) was applied to determine differences between two means. For the comparison of multiple groups, two-way ANOVA test was used. For all statistical analyses, significance values are indicated as NS ($p > 0.05$), (*$p \leq 0.05$), (**$p \leq 0.01$), (***$p \leq 0.001$)).

## Reporting summary

Further information on research design is available in the Nature Portfolio Reporting Summary linked to this article.

## Data availability

The authors declare that all relevant data are included in the main text and/or its supplementary information files. All other data are available from the corresponding author on reasonable request. Numerical source data for all figures can be found in Supplementary Data. Unprocessed gels can be found in Supplementary Fig. 14 with Image Lab 6.1.

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

## Acknowledgements

The bulk and single cell rna_seq data were analyzed using the R studio. The current study was supported by grants from the National Natural Science Foundation of China (Grant No. 81803064), the Natural Science Foundation of Jiangxi (Grant No. 20192ACB21027), the Natural Science Foundation of Guangdong (Grant No. 2022A1515140178) and Doctorate Foundation of Guangdong Medical University (Grant No. 1026/4SG22195G). Part works were performed in the Dr. Weiping Min's lab at Nanchang University.

## Author contributions

All the authors are recognized to contribute to this manuscript. Y.Z. planned experiments; H.R., W.H., Y.Z., H.X., W.H., N.H., and C.Z. performed experiments; Y.Z. analyzed the single cell data and TCGA database; Y.Z. wrote the paper. All authors are assured that we met the criteria for authorship and all reviewed the manuscript.

## Competing interests

The authors declare no competing interests.
