## [Peer Review File · Communications Biology]

Reviewers' comments:

Reviewer #1 (Remarks to the Author):

This manuscript is titled "Tumor-targeted nanodrug FSGG/siGal-9 for transdermal photothermal immunotherapy of melanoma" by Huihong Ren et al. used the photothermal conversion abilities of nanoparticles against melanoma.

Comments:

The authors should explain why in vitro the 808 NIR laser was used at 2 W/cm² but in vivo at 1W/cm² ? also, I am not clear about the time exposure for both cases.

In preparation of GNR, the authors must state how long they let the seed solution sit before adding to the growth solution.

The authors must show the plasmon bands of GNRs before and after adding the FSG occur in the NIR-IR /biological optical window, I couldn't find any supporting information . What was the aspect ratio of GNR?

Reviewer #2 (Remarks to the Author):

In the manuscript entitled: "Tumor-targeted nanodrug FSGG/siGal-9 for transdermal photothermal immunotherapy of melanoma", Zhang et al. describe the use of photothermal nanosensitizer (Folic acid-modified and gold nanorod-linked iron oxide/silica core/shell nanostructure, FSGG) loaded with Gal-9 siRNA (FSGG/siGal-9) for targeting B16F10 cells and block the Gal-9/TIM-3 axis and prohibit T cell exhaustion.

Strengths of this pre-clinical translational manuscript include a relative novel cargo, Galectin-9 (Gal-9) siRNA, attached to a gold-based nanorod-linked structure, and strong renowned scientists.

Unfortunately, the rationale of targeting Gal-9 in melanoma is not clear/strong. The genomic data provided in panels of Figure 2 show no upregulation of Gal-9 in melanoma cells vs. normal cell and/or an increase with tumor stage. However, literature shows that there is a differential expression of Gal-9 on various other tumor types and diseases. Additionally, Gal-9 is expressed on non-tumor cells as well, which raises the question of the feasibility as an anti-cancer treatment in the clinic. Similarly, the gold nanorods aggregate (Figure 1), as many nanoparticles do, which is also be a concern developing it further. Figure 3 is not well described and its overall function is not clear. It essentially shows the stimulating properties of B16F10 derived exosomes. The suggested correlation that T cell activation is mediated through Gal-9 is circumstantial and siGal-9 exosomes seem to induce T cell activation as well. Panel 3E is illegible printed. The data presented in Figure 4 seems more appropriate as supplemental material as it shows no toxicity in any of the systemic parameters tested when applying FSGG topically without siRNA or laser – essential for the PTT ability. These controls are missing. Similarly, one wonders about the stability of siRNA after 5 minutes exposure of 1-5 W/cm². It is unclear why in Figure 6 the attention shifts to induction of apoptosis of Gal-9 expression DC and T cells in the spleen, as the rationale was to target Gal-9 on B16F10 cells. Additionally, why is sometimes the MTT assay used and other times the CCK-8 or LDH assay. The latter ones are not mentioned in the Materials and Method section. It is unclear from the description whether the authors fully appreciate the differences between cell viability and proliferation. It is unclear whether the mice were shaven prior to the various interventions, esp. in light of topical treatments. Similarly, the

incubating B16F10 cell with T cells at ratios 1-50 through 1:200 and using LDH in the supernatant is contrived. INF γ staining would have been more insightful as a readout. Throughout it is unclear if the effects seen are only B16F10 specific. Very minor: some typos and some syntax mistakes throughout the manuscript.

Point-by-point response to the referees' comments.

Reviewers' comments:

Reviewer #1 (Remarks to the Author):

This manuscript is titled "Tumor-targeted nanodrug FSGG/siGal-9 for transdermal photothermal immunotherapy of melanoma" by Huihong Ren et al. used the photothermal conversion abilities of nanoparticles against melanoma.

Comments:

The authors should explain why in vitro the 808 NIR laser was used at 2 W/cm² but in vivo at 1 W/cm²? also, I am not clear about the time exposure for both cases.

Response: Thank you for your comments! We also have tested the 2 W/cm² in vivo elsewhere, but here we used the 1 W/cm² for the combination treatment with siGal-9. We prolonged the exposure time to 500 sec in vitro and used 300 sec (5 min) for the in vivo treatment.

In preparation of GNR, the authors must state how long they let the seed solution sit before adding to the growth solution.

Response: Thank you for your questing. We have let the seed solution sit for 2 hrs before use, which also have been added and marked in the manuscript (Page 6).

The authors must show the plasmon bands of GNRs before and after adding the FSG occur in the NIR-IR /biological optical window, I couldn't find any supporting information. What was the aspect ratio of GNR?

Response: Those plasmon bands of GNR-MUA, Fe₃O₄@SiO₂ (FS), Fe₃O₄@SiO₂-Glu (FSG) and Fe₃O₄@SiO₂-GNR-Glu (FSGG) were provided in Figure 1(D) using UV-Vis absorption spectra, and the synthesis step is shown in the following (Page 8):

The appearance of GNR-MUA was characterized using TEM as shown in the Supplementary Figure 1 and the aspect ratio was calculated about 3.5 that was added and marked in the manuscript (Page 17).

Reviewer #2 (Remarks to the Author):

In the manuscript entitled: "Tumor-targeted nanodrug FSGG/siGal-9 for transdermal photothermal immunotherapy of melanoma", Zhang et al. describe the use of photothermal nanosensitizer (Folic acid-modified and gold nanorod-linked iron oxide/silica core/shell nanostructure, FSGG) loaded with Gal-9 siRNA (FSGG/siGal-9) for targeting B16F10 cells and block the Gal-9/TIM-3 axis and prohibit T cell exhaustion.

Strengths of this pre-clinical translational manuscript include a relative novel cargo, Galectin-9 (Gal-9) siRNA, attached to a gold-based nanorod-linked structure, and strong renowned scientists.

Comments:

Unfortunately, the rationale of targeting Gal-9 in melanoma is not clear/strong. The genomic data provided in panels of Figure 2 show no upregulation of Gal-9 in melanoma cells vs. normal cell and/or an increase with tumor stage. However, literature shows that there is a differential expression of Gal-9 on various other tumor types and diseases. Additionally, Gal-9 is expressed on non-tumor cells as well, which raises the question of the feasibility as an anti-cancer treatment in the clinic.

Response: We calculated the expression levels of Gal-9 and related PD-L1 and CD28, and compared the expression levels of Gal-9 between tumor vs normal tissue, and also checked the relationship of expression level of Gal-9 with tumor stage in the top ten type tumors from TCGA database for answering this question. The results are shown in Supplementary Figure 2-12. The expression levels of Gal-9 are much higher than that of PD-L1 and CD28 in tumors (Supplementary Figure 2), combining that we observed Gal-9 may be the main reason to induce effector T cells exhaustion and also immune tolerance, which indicated more feasibility as anti-cancer treatment in the clinic. There are significantly higher levels in four type tumors (BRCA, KIRC, THCA, UCEC) and significant lower levels in two type tumors (PRAD, LUSC), and no remarkable differential expression of Gal-9 with the tumor stages as shown in the Supplementary Figure 3-12.

Similarly, the gold nanorods aggregate (Figure 1), as many nanoparticles do, which is also be a concern developing it further.

Response: Thanks for reviewer's kindly suggestion. Yes, we will develop it further.

Figure 3 is not well described and its overall function is not clear. It essentially shows the stimulating properties of B16F10 derived exosomes. The suggested correlation that T cell activation is mediated through Gal-9 is circumstantial and siGal-9 exosomes seem to induce T cell activation as well. Panel 3E is illegible printed.

Response: Yes, what the reviewer mentioned is that we want to show here. This is a part of results for this project and so we showed. Panel 3E is repainted as well.

The data presented in Figure 4 seems more appropriate as supplemental material as it shows no toxicity in any of the systemic parameters tested when applying FSGG topically without siRNA or laser – essential for the PTT ability. These controls are missing.

Response: Since the toxicities of drugs and drug carriers are important, so we played this results in the manuscript. The control for the TEM results (Figure 4D) are shown in the Supplementary Figure 14 with different magnifications. We donot observed any histopathological and functional abnormalities in any of the mouse organs, and the accumulation of materials and abnormalities of cellular structures in the liver and spleen.

Similarly, one wonders about the stability of siRNA after 5 minutes exposure of 1-5 W/cm².

Response: Yes, we didnot test the stability of siRNA after 5 minutes exposure of 1-5 W/cm², but it could effectively silence Gal9 in vivo after 5 minutes exposure of 1W/cm² (Figure 5E).

It is unclear why in Figure 6 the attention shifts to induction of apoptosis of Gal-9 expression DC and T cells in the spleen, as the rationale was to target Gal-9 on B16F10 cells.

Response: We tested the expression levels of GAL9 on DC as the important APC to activate T cell immune response for indicating changes of immune system status. The apoptosis of T cells exhibits their biological activity also for indicating changes of immune system status.

Additionally, why is sometimes the MTT assay used and other times the CCK-8 or LDH assay. The latter ones are not mentioned in the Materials and Method section. It is unclear from the description whether the authors fully appreciate the differences between cell viability and proliferation.

Response: We applied the MTT assay testing the cell viability in Figures 1(I and J), the CCK-8 assay testing the proliferation of cells in Figures 3(D) and 6(E), and the LDH assay testing the cytotoxicity in Figure 6(F). The experiment parameters were provided and marked with yellow in Materials and Methods (Page 9, 10, 15 and 16).

- 1) The MTT assay: Thiazolyl Blue is easily transferred into plasma from the medium and then reduced to insoluble formazan by NAD (P) H Oxidoreductase in cells. It is often used for measurement of living cells.
- 2) The CCK8 assay: WST-8 can be reduced by dehydrogenase in mitochondria to produce highly water-soluble orange yellow formazan products that is released into the culture medium. It is often used to detect cell proliferation.
- 3) The LDH assay detects the activity of lactate dehydrogenase (LDH) in the culture medium. After cells such as tumor cells are killed by drugs, LDH can be released into medium for testing the killing ability of drugs.

Thiazolyl Blue

WST-8

It is unclear whether the mice were shaven prior to the various interventions, esp. in light of topical treatments.

Response: Thanks. We have corrected this inattention and marked the insertions in the manuscript.

Similarly, the incubating B16F10 cell with T cells at ratios 1-50 through 1:200 and using LDH in the supernatant is contrived. INF γ staining would have been more insightful as a readout.

Response: The killing mechanisms of CTL to target tumor cells: 1) Release Perforin-1 and Granzyme to kill target cells, 2) Induce apoptosis of target cells mediated by FasL, INF γ and TNF. Therefore, we tested LDH in the supernatant. We will try to use the INF γ staining assessing the killing capacity in the future.

Throughout it is unclear if the effects seen are only B16F10 specific.

Response: Thanks. We will test the effects on more types of cancer cells in the future.

Very minor: some typos and some syntax mistakes throughout the manuscript.

Response: Thanks. We have checked the whole manuscript and done the corrections.

Reviewers' comments:

Reviewer #3 (Remarks to the Author):

The premise of this paper is promising, combination photo-immunotherapy. There is some rich and beautiful data in the paper, but it needs a major re-writing before it can be considered further. Some suggestions for its improvement prior to resubmission are as follows:

- 1) Abstract: at present the abstract is too general and does not summarize the data,
- 2) Introduction: As often happens when one does innovative work, it can be difficult to weave a convincing background because of lack of work that is similar to the current. That said, for this paper, one would want to write an introduction that lays out the premise of this work. For example, WHY would combining these modalities be more effective and how did the authors arrive at their unique approach. The current version falls short of this.
- 3) Results: The results need to be better organized. Many figures have too many panels and don't stand out or its unclear what they are. Instead of schematically showing what the nanoparticles should be, maybe Fig. 1 should focus only on characterization. To me Fig. 2 maybe seems like target validation? Whereas Fig. 3 is more about effects on the target? In essence the results are difficult to follow and don't follow a logical flow. Tox and efficacy studies are important and these figures do stand alone.

Basically I think this paper has a lot of merit but the authors almost need to start from scratch in re-writing it. Let the results tell the story, so re-organize the data to show an argument which builds figure after figure. At present the organization of the paper strongly distracts or is confusing and isn't painting the data in its best light.

Point-by-point response to the referees' comments.

Reviewers' comments: Reviewer #3 (Remarks to the Author): The premise of this paper is promising, combination photo-immunotherapy. There is some rich and beautiful data in the paper, but it needs a major re-writing before it can be considered further. Some suggestions for its improvement prior to resubmission are as follows:

1) Abstract: at present the abstract is too general and does not summarize the data,

Response: We have added new data and rewritten the summary in the abstract.

2) Introduction: As often happens when one does innovative work, it can be difficult to weave a convincing background because of lack of work that is similar to the current. That said, for this paper, one would want to write an introduction that lays out the premise of this work. For example, WHY would combining these modalities be more effective and how did the authors arrive at their unique approach. The current version falls short of this.

Response: We have added more backgrounds for explaining this work in the introduction, and done more works marked in yellow color in the Results.

3) Results: The results need to be better organized. Many figures have too many panels and don't stand out or it's unclear what they are. Instead of schematically showing what the nanoparticles should be, maybe Fig. 1 should focus only on characterization. To me Fig. 2 maybe seems like target validation? Whereas Fig. 3 is more about effects on the target? In essence the results are difficult to follow and don't follow a logical flow. Tox and efficacy studies are important and these figures do stand alone.

Basically I think this paper has a lot of merit but the authors almost need to start from scratch in re-writing it. Let the results tell the story, so re-organize the data to show an argument which builds figure after figure. At present the organization of the paper strongly distracts or is confusing and isn't painting the data in its best light.

Response: Thanks a lot, we have carefully considered the pointed-out shortcomings, and reorganized the whole paper.